# Hyperspectral Unmixing with Gaussian Mixture Model and Spatial Group Sparsity

**Qiwen Jin** [1] , **Yong Ma** [1,2], **Erting Pan** [1], **Fan Fan** [1,2], **Jun Huang** [1,2], **Hao Li** [3] , **Chenhong Sui** [4] **and Xiaoguang Mei** [1,2,*]

1   Electronic Information School, Wuhan University, Wuhan 430072, China; fruitsprince@whu.edu.cn (Q.J.); mayong@whu.edu.cn (Y.M.); panerting@whu.edu.cn (E.P.); fanfan@whu.edu.cn (F.F.); junhwong@whu.edu.cn (J.H.)
2   Institute of Aerospace Science and Technology, Wuhan University, Whan 430079, China
3   College of Mathematics and Computer Science, Wuhan Polytechnic University, Wuhan 430023, China; lihao@whpu.edu.cn
4   School of Optoelectronic Information Science and Technology, Yantai University, Yantai 264003, China; sui6662015@ytu.edu.cn
*   Correspondence: meixiaoguang@whu.edu.cn

**Abstract:** In recent years, endmember variability has received much attention in the field of hyperspectral unmixing. To solve the problem caused by the inaccuracy of the endmember signature, the endmembers are usually modeled to assume followed by a statistical distribution. However, those distribution-based methods only use the spectral information alone and do not fully exploit the possible local spatial correlation. When the pixels lie on the inhomogeneous region, the abundances of the neighboring pixels will not share the same prior constraints. Thus, in this paper, to achieve better abundance estimation performance, a method based on the Gaussian mixture model (GMM) and spatial group sparsity constraint is proposed. To fully exploit the group structure, we take the superpixel segmentation (SS) as preprocessing to generate the spatial groups. Then, we use GMM to model the endmember distribution, incorporating the spatial group sparsity as a mixed-norm regularization into the objective function. Finally, under the Bayesian framework, the conditional density function leads to a standard maximum a posteriori (MAP) problem, which can be solved using generalized expectation-maximization (GEM). Experiments on simulated and real hyperspectral data demonstrate that the proposed algorithm has higher unmixing precision compared with other state-of-the-art methods.

**Keywords:** hyperspectral unmixing; Gaussian mixture model; spatial group sparsity; superpixel segmentation; endmember variability; Bayesian framework

## 1. Introduction

Over the past few decades, the rich spatial–spectral joint information of hyperspectral imaging (HSI) has greatly improved the sensing ability of remote sensing images. With the unique advantages of high spectral resolution and union of imagery and spectrum, HSI has been widely used in various fields, such as agricultural remote sensing, object classification, environment monitoring, and military investigation [1–8]. However, limited by the spatial resolution of the instrument, atmospheric mixing effects, and the complex natural surface, a single pixel always contains more than one spectrum of features, resulting in a "mixed pixel" phenomenon. This brings great difficulty to the accurate interpretation and recognition of hyperspectral image contents. Thus, the spectral unmixing (SU), which refers to identify the proportion (abundance) of the basic constituent spectra (endmembers) in

mixed pixels at the subpixel level, has been a major issue in hyperspectral remote sensing applications, and has recently been extensively investigated [9–13].

The linear mixture model (LMM) is often used to describe the mixed pixel phenomenon due to its simplicity and certain physical meaning, which assumes that the observed spectra are linearly combined by the endmembers according to their respective abundance coefficients. The mathematical form of LMM can be expressed as

$$\mathbf{y}_n = \sum_{j=1}^{M} \mathbf{m}_j \alpha_{nj} + \mathbf{n}_n, \text{ s.t. } \alpha_{nj} \geq 0, \sum_{j=1}^{M} \alpha_{nj} = 1, \tag{1}$$

where $\mathbf{y}_n \in \mathcal{R}^B, n = 1, ...N$ is the observed spectra ($N$ denotes the number of pixels; $B$ denotes the number of bands), $\mathbf{m}_j$ is the pure material (called *endmember*), $\alpha_{nj}$ is the corresponding proportions (called *abundance*), and $\mathbf{n}_n \in \mathbb{R}^B$ is additive noise. We have to note that the endmember set $\mathbf{m}_j : j = 1, ..., M$ is fixed for all the pixels. In addition, there are two physical abundance constraints need to be considered in linear unmixing, the so-called abundance non-negative constraints (ANC) and the abundance sum-to-one constraints (ASC).

In reality, due to environmental, atmospheric, and temporal factors, the spectrum of the same pure substance received by the detector tends to change greatly. Even for the pure pixels that only contain one material, the spectrum of the entire image may not be the same. The inaccuracy of the endmember signature will affect the performance of the unmixing to a great extent. Therefore, in practical applications, the unmixing process must solve the interference caused by the endmember variability. In general, the mathematical form by considering the endmember variability can be expressed a

$$\mathbf{y}_n = \sum_{j=1}^{M} \mathbf{m}_{nj} \alpha_{nj} + \mathbf{n}_n, \text{ s.t. } \alpha_{nj} \geq 0, \sum_{j=1}^{M} \alpha_{nj} = 1, \tag{2}$$

where $\mathbf{m}_{nj} \in \mathbb{R}^B : j = 1, ..., M, n = 1, ..., N$. Here, the endmember set is not fixed, and the endmember spectrum of each pixel $\mathbf{y}_n$ could be different. Compared with the fixed endmember set, the use of multiple endmembers per class has a more interpretable physical meaning and higher accuracy in endmember and abundance estimation.

In recent years, many approaches and techniques have been proposed to solve the problem caused by the endmember variability. In [14,15], the authors reviewed methods that take into account the endmember variability can be expressed in two categories: (1) endmembers represented as a discrete set or (2) endmembers represented as a continuous distribution. In the first category, the spectrum corresponding to each pixel is supposed to be represented as a convex combination of elements within the set. The core of this method is selecting the one with the smallest error by trying every endmember combination, such as multiple endmember spectral mixture analysis (MESMA) [16], AutoMCU [17,18], and BSMA [19]. However, those methods mentioned based on the endmember set have a common disadvantage: when the spectral library and combination of the collection elements is large, their complexity will increase exponentially and resulting in extreme searching efficiency problem.

The second category usually takes the statistical method to model the distribution of endmembers. The core of the method is to assume that the endmember of each pixel is derived from a sample of a certain probability distribution. This method of modeling probabilistic sampling implies embracing the large libraries, and the numerical values of the model are solvable. The current popular method based on this category includes normal compositional model (NCM) [20], Beta compositional model (BCM) [21], and Gaussian Mixture model (GMM) [22,23]. As NCM and BCM are both based on the unimodal center hypothesis, in actual scenarios, the distribution of the endmember may not be well represented. GMM takes a mixture of Gaussians to approximate any distribution that

endmembers $\mathbf{m}_{nj}$ may exhibit. Therefore, it has better flexibility accuracy in solving the problem of endmember variability.

The performance of those methods in the second category often has a significant dependence on the initial value of parameters but does not rely on the large-scale spectral database. Therefore, the model is relatively intuitive and has been a research hotspot for the endmember variability problem. However, the methods mentioned above only use spectral information alone and do not make full use of the spatial information in the scene, because pixels are only treated as isolated entities without considering the local correlation between them. The authors of [8,24–27] proved that the spatial correlation between the observed pixels can be incorporated into in the unmixing process, which can further enhance the performance in both the endmember extraction and abundance estimation. In that vein, Iordache et al. [28] incorporated a spatial constraint in sparse unmixing by the total variation regularizer, to find the minimum difference of the abundance among the neighboring pixels and promote the piecewise smoothness. Zhou et al. [22] used the Laplacian smoothness and sparsity constraint as the prior knowledge in GMM models to improve the accuracy of the abundance estimation. In [26], a low-rank representation (LRR) method for HSI unmixing is proposed to find a low-rank property in abundance maps and utilize spatial information between local pixels.

The above methods are dedicated to fully exploiting spatial and spectral information. Nevertheless, the prior knowledge or constraints assumption requires a rather strict condition that the abundance of local pixels should be piecewise smooth, which means that the associated abundance of the mixed pixels and their neighboring pixels should be similar. When the pixels are located at the boundaries of different materials or the inhomogeneous region, the abundance in adjacent pixels does not have these prior constraints. Thus, the local prior knowledge is not applicable in the whole HSI scene and numerous methods based on spatial segmentation preprocessing have been proposed. In [29], an iterative process with different sliding windows for the automated morphological endmember extraction was proposed, which is one of the famous algorithms incorporating local spatial information. Liu et al. incorporated an abundance separation and smoothness constraints on the spectral and spatial domains respectively to promote the unmixing results [30]. However, those methods use a fixed-size rectangle strategy. When the size of the maximum sliding window is relatively large, this will be a time-consuming process. When the partitioned regions are fixed, the shape and size can not be accustomed on the basis of different spatial structures. Recently, Wang et al. [31] proposed a spatial group sparsity regularized non-negative matrix factorization (SGSNMF) method by minimizing the reconstruction error. This algorithm takes superpixel segmentation (SS) as processing and integrates the spatial group sparsity as group-structured prior information, and obtains relatively high efficiency and precision in HSI unmixing.

Thus, in this paper, in an attempt to solve the problems caused by endmember variability, fully consider the possible spatial correlation between local pixels, and achieve better abundance estimation performance, a spatial group sparsity constraint based on Gaussian mixture model (SGSGMM) is proposed. Motivated by the mentioned problems, first, we adopt the adaptive size superpixel strategy to customize the shape and size of the unmixing region based on different spatial structures and considering each superpixel as a nonoverlapping region. In these regions, pixels are highly spatial correlated, the mixing pixel and its associated abundance should share the same prior probability. Secondly, we incorporate the spatial group sparsity constraints in each superpixel as a prior knowledge into the objective function. Finally, considering the endmember variability phenomenon, we use GMM to model the endmember distribution. Under the Bayesian framework, the conditional density function leads to a standard maximum a posteriori (MAP) problem, and a generalized expectation-maximization is obtained to solve the objective function [32]. Compared with other popular algorithms on both simulated and real hyperspectral datasets, the proposed method SGSGMM can perform more accurate unmixing results.

The rest of this paper is structured as follows. The related GMM methods are briefly introduced in Section 2. The details of the proposed model SGSGMM and the GEM for solving and optimization

are introduced in Section 3. The performances of the proposed SGSGMM and comparison with other state-of-art algorithms in synthetic datasets and real HSIs are presented in Section 4. The discussions and conclusions of this paper are introduced in Sections 5 and 6.

## 2. Related Models

As the proposed model of this paper is based on the GMM framework, we will introduce the formulation of GMM in the following section briefly.

The GMM method [22] is an LMM that considers the endmember variability and uses a mixture of Gaussians to approximate any distribution of the endmember. The method can be classified in the second category mentioned above (endmembers represented using a continuous distribution) and starts by assuming the endmember $p(\mathbf{m}_{nj})$ follows a mixture of Gaussians distribution. The density function $p(\mathbf{m}_{nj})$ can be described as

$$p(\mathbf{m}_{nj}|\mathbf{\Theta}) := f_{\mathbf{m}_j}(\mathbf{m}_{nj}) = \sum_{k=1}^{K_j} \pi_{jk} \mathcal{N}(\mathbf{m}_{nj}|\boldsymbol{\mu}_{jk}, \boldsymbol{\Sigma}_{jk}), \tag{3}$$

s.t. $\pi_{jk} \geq 0, \sum_{k=1}^{K_j} \pi_{jk} = 1$, with $K_j$ being the number of components; $\pi_{jk}$ denotes the weight of its $k$th Gaussian component in $j$th endmember, $\boldsymbol{\mu}_{jk} \in \mathbb{R}^B$; and $\boldsymbol{\Sigma}_{jk} \in \mathbb{R}^{B \times B}$ denotes the mean matrix and covariance matrix, respectively. Here, $\mathbf{\Theta} := \pi_{jk}, \boldsymbol{\mu}_{jk}, \boldsymbol{\Sigma}_{jk} : j = 1, ..., M, k = 1, ..., K_j, \mathbf{m}_{nj} : j = 1, ..., M$ are independent to each other.

The noise $\mathbf{n}_n$ is also assumed to follow the Gaussian distribution, which has the density function $p(\mathbf{n}_n) := \mathcal{N}(\mathbf{n}_n|0, \mathbf{D})$, where $\mathbf{D}$ denotes covariance matrix of noise, obtains the distribution of $\mathbf{y}_n$ by $\mathbf{y}_n = \sum_{j=1}^{M} \mathbf{m}_{nj} \alpha_{nj} + \mathbf{n}_n$, the observed pixel data $\mathbf{y}_n$ will be another mixture of Gaussians, and we can obtain the density function of $\mathbf{y}_n$ as

$$p(\mathbf{y}_n|\boldsymbol{\alpha}_n, \mathbf{\Theta}, \mathbf{D}) = \sum_{\mathbf{k} \in \mathcal{K}} \pi_{\mathbf{k}} \mathcal{N}(\mathbf{y}_n|\boldsymbol{\mu}_{n\mathbf{k}}, \boldsymbol{\Sigma}_{n\mathbf{k}}), \tag{4}$$

where $\alpha_n := [\alpha_{n1}, ... \alpha_{nM}]^T$, $\mathcal{K} := \{1, ..., K_1\} \times \{1, ..., K_2\} \times ... \times \{1, ..., K_M\}$ is the Cartesian product of the M index sets, $\mathbf{k} := (k_1, ..., k_M) \in \mathcal{K}, \pi_{\mathbf{k}} \in \mathbb{R}, \boldsymbol{\Sigma}_{n\mathbf{k}} \in \mathbb{R}^{B \times B}$ are defined by

$$\pi_{\mathbf{k}} := \prod_{j=1}^{M} \pi_{jk_j}, \boldsymbol{\mu}_{n\mathbf{k}} := \sum_{j=1}^{M} \alpha_{nj} \boldsymbol{\mu}_{jk_j}, \boldsymbol{\Sigma}_{n\mathbf{k}} := \sum_{j=1}^{M} \alpha_{nj}^2 \boldsymbol{\Sigma}_{jk_j} + \mathbf{D}. \tag{5}$$

Here, if we set $\mathcal{K} := \{1\} \times \{1\} \times ... \times \{1\}$, then we have the weight of the Gaussian component $\pi_k = \pi_{11} = ... \pi_{M1} = 1$, which will exactly become the formulation of the NCM model. In that case, the distribution of the mixed pixel $\mathbf{y}_n$ becomes

$$p(\mathbf{y}_n|\boldsymbol{\alpha}_n, \mathbf{\Theta}, \mathbf{D}) = \mathcal{N}\left(\mathbf{y}_n| \sum_{j=1}^{M} \alpha_{nj} \boldsymbol{\mu}_{j1}, \sum_{j=1}^{M} \alpha_{nj}^2 \boldsymbol{\Sigma}_{j1} + \mathbf{D}\right) \tag{6}$$

where $\mathbf{\Theta} := \{\boldsymbol{\mu}_j, \boldsymbol{\Sigma}_j := j = 1, ..., M\}$. We can clearly find from the above equation that the probability distribution of each endmember is only one Gaussian component. Here, we intuitively introduce how the GMM and NCM models work and the difference between them. NCM minimizes the error with $\mathbf{y}_n$ in the LMM linear combination by finding the Gaussian center $\boldsymbol{\mu}_j$ of each endmember, and the error weights can be seen balanced by its covariance matrix. GMM minimizes the error with $\mathbf{y}_n$ in the LMM linear combination by finding a group of Gaussian centers $\boldsymbol{\mu}_{jk}$ of each endmember, and the error weighted is balanced by the prior $\pi_k$. Although NCM can be much improved in terms of computational complexity compared to GMM, the single-peak hypothesis could not approximate the endmember probability distribution in real scenes.

## 3. GMM Unmixing with Superpixel Segmentation and Spatial Group Sparsity

In this section, first, we describe the specific steps in implementing the superpixel segmentation, then introduce the GMM unmixing based on the spatial group sparsity. Finally, the details of using the GEM algorithm to solve SGSGMM are described.

### 3.1. Formulation of the Proposed SGSGMM

The main goal of this paper is to solve the problem of endmember variability, taking into account the possible spatial correlation between local pixels and seeking proper prior constraints to achieve better abundance estimation. Endmember variability makes the spectrum of the same material appear different in the spectral domain. The methods solving this problem can be expressed in two categories: one is the endmember set, and the other is the continuous distribution. The methods of the first category both try to search the minimum mean square error through all the endmember combination. Thus, they will both face with the computational inefficiency difficulties. The second category takes the statistical analysis of the pure pixels and assumes following a certain form of distribution. The current popular unmixing methods of the second category have also been mentioned above: Normal compositional model (NCM) [20], Beta compositional model (BCM) [21], and Gaussian mixture model (GMM) [22]. Nevertheless, they all ignore the local spatial correlation of HSI. More specifically, to incorporate the spatial correlations between the observed pixels into the unmixing process. Zhou et al. [22] assumes the abundances **A** have the proper smoothness and sparsity prior constraints. These prior constraints have enhanced the performance of abundance estimation to a great extent. However, those constraints both require a rather strict assumption that the associated abundance of the mixed pixel should be similar, or the pixel are located in the homogeneous regions. When the pixels' region is not pure or lies on the boundaries of different materials, the abundance in adjacent pixels will not have any sparse or smooth prior constraints. Thus, taking prior knowledge into the whole HSI scene is not applicable under the unmixing process. In that vein, to cut the HSI into several homogeneous regions, so that the pixels in the same superpixels incline to have common features and a high spatial correlation, can better incorporate the abundance prior constraints.

From another perspective, the first step of the GMM approach is to separate the library into $M$ groups, where each group represents a material and is clustered into several centers. Then the endmember combinations take place by picking one center from each group. Therefore, in the GMM method, the size of each cluster will influence the probability of selecting its center to a large degree. When the original HSI is difficult to separate into several clusters, directly using the GMM method for unmixing causes the endmember cluster to not fit the ground truth very well. Sometimes GMM even failed to estimated the distribution of clusters, and influence the unmixing accuracy of the abundance. This case will be introduced in later experiments. Thus, taking SS to segment HSI will help the GMM to separate the clusters better, and reduce the risk of failing to estimated the clusters distribution. In this paper, we adopt the modified simple linear iterative clustering (SLIC) [31] algorithm to cut the HSI into different regions. Compared with the original SLIC [33], the shape of each superpixel of the method is composed of a regular hexagon. The advantage over the original SLIC is that each hexagon has more nondiagonal neighborhoods than squares. Therefore, we can get more homogeneous spatial regions as the center of initialization.

As shown in Figure 1, here, we take Principal Component Analysis (PCA) to obtain the first principle component of HSI, which is used as the base image when conducting the SS. Then, the original HSI is separated into several nonoverlapping and homogeneous regions, which are accustomed based on different spatial structures. Therefore, the pixels are highly spatially correlated. Due to the spatial dependence, each spatial group is expected to share the same endmember assignment and structure property. Furthermore, in each superpixel, only a few corresponding endmembers will participate in the process of unmixing, which means the shared structure property should be sparse. To take into account both spatial and sparse priors of the abundance, we should seek a more appropriate sparsity constraint at the level of spatial groups rather than the whole image. Recently, several mixed-norm

regularizations have been proposed in machine learning, computer vision, and statistics [34,35]. One typical example is Li et al., in which a $l_{2,1}$-norm regularizer $f(\mathbf{A}) = \sum_{i=1}^{\hat{M}} ||\mathbf{A}^i||_2$ is taken to exploit the group row sparsity within the abundance matrix $\mathbf{A}$. Similarly, Yang et al. proposed the spatial group sparse coding (SGSC) by extending the robust ability among the group training regions [36]. Here, we adopt the spatial group sparsity constraints, which is proposed in SGSNMF [31], which is generalized as

$$f(\mathbf{A}) = \sum_{p=1}^{P} \sum_{\mathbf{A}_j \in \vartheta_p} c_j ||\mathbf{W}_p \mathbf{A}_j||_2, \tag{7}$$

where $\vartheta_p$ is the spatial groups (superpixels), $P$ is the number of spatial groups, and $c_j$ is the pixel-by-superpixel confidence index, which is defined as the inversely proportional to the spatial–spectral distance; $\mathbf{D}_j^P$, $\mathbf{W}^p = \text{diag}(\mathbf{W}_1^p, ... \mathbf{W}_M^p) \in \mathcal{R}^{M \times M}$ is a diagonal matrix. The mathematical form can be expressed as

$$c_j = \frac{1}{\mathbf{D}_j^P} \tag{8}$$

$$\mathbf{W}_i^p = \frac{1}{|\bar{\mathbf{A}}^p[i]| + \varepsilon} \tag{9}$$

where $\bar{\mathbf{A}}^p = [\bar{\mathbf{A}}^p[1], ..., \bar{\mathbf{A}}^p[M]]^T$ is the average abundance vector of the $p$th superpixel. Therefore, the $\mathbf{W}_i^p$ matrix is equivalent to a weight matrix to prevent loss of spatial details inside each superpixel. $\varepsilon$ is an extremely small number to prevent the $\mathbf{W}_i^p$ matrix from approaching infinity when the average abundance within the superpixel is zero.

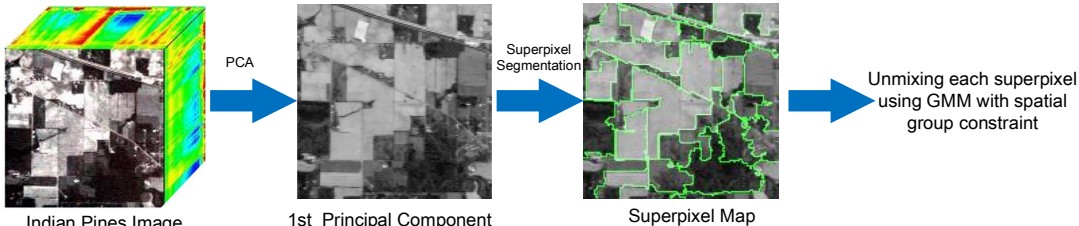

**Figure 1.** Algorithm flow chart of the proposed spatial group sparsity constraint based on Gaussian mixture model (SGSGMM) method.

In the original SGSNMF method, the spatial distance and spectral distance is measured by the Euclidean distance (ED) and spectral angle distance (SAD), respectively. Here, we adopt the SID-SAM method to measure the spectral distance [37]; because it can consider the spectral angle and information divergence of pixels at the same time, and, in the specific experiment, it can better capture the variability and similarity between different pixels. In terms of spatial distance, we also use the ED method as the standard for measurement. The mathematical form can be expressed as

$$\begin{aligned}\mathbf{D}_{\text{spectral}} &= \text{SID}(x_i, x_c) \times \tan(\text{SAM}(x_i, x_c)), \\ \mathbf{D}_{\text{spatial}} &= \frac{\sqrt{(ix - cx)^2 + (iy - cy)^2}}{r}\end{aligned} \tag{10}$$

where $\mathbf{D}_{\text{spectral}}$ and $\mathbf{D}_{\text{spatial}}$ denote the spatial and space distance, respectively. $x_i$ denotes $i$th pixel of the image, and $x_c$ denotes the cluster center of the $c$th superpixel. $(ix, iy), (cx, cy)$ denote the spatial coordinates of the $i$th pixel and cluster center of the $c$th superpixel. $r$ is the average size of the superpixels, which is utilized to control the total number of spatial groups. SID and SAM measurement are defined as

$$
\begin{aligned}
\mathrm{SID}(x_i, x_c) &= D(x_i||x_c) + D(x_c||x_i) \\
&= \sum_{k=1}^{B} p_{ik} \log\left(\frac{p_{ik}}{p_{ck}}\right) + \sum_{k=1}^{B} p_{ck} \log\left(\frac{p_{ck}}{p_{ik}}\right) \\
&= \sum_{k=1}^{B} (p_{ik} - p_{ck}) \log\left(\frac{p_{ik}}{p_{ck}}\right)
\end{aligned}
\tag{11}
$$

$$
\mathrm{SAM}(x_c, x_c) = \cos^{-1}\left(\frac{<x_i, x_c>}{||x_i||\,||x_c||}\right)
\tag{12}
$$

where $p_{ik} = \frac{x_{ik}}{\sum_{k=1}^{B} x_{ik}}$, $p_{ck} = \frac{x_{ck}}{\sum_{k=1}^{B} x_{ck}}$ and $x_{ik}$ denotes the $k$th band of pixel $x_i$.

Besides, for improved weighting of the relative importance between spatial and spectral similarities, we add the parameter $\lambda$ to balance the spectral and spatial items. Then, the spatial–spectral distance $\mathbf{D}_j$ can be generated as

$$
\mathbf{D}_j = (1 - \lambda) \times \mathbf{D}_{\text{spectral}} + \lambda \times \mathbf{D}_{\text{spatial}},
\tag{13}
$$

Furthermore, considering that in the same superpixel, adjacent pixels are more likely to be composed of the same material and thus have similarities. Therefore, combining proper smoothness and sparsity prior constraints, the density function of the abundances $\mathbf{A}$ can be generalized as

$$
p(\mathbf{A}) \propto \exp\left\{ \sum_{p=1}^{P}\left(-\frac{\beta_1}{2}\mathrm{Tr}\left(\mathbf{A}^T\mathbf{L}\mathbf{A}\right) + \frac{\beta_2}{2}\sum_{\mathbf{A}_j \in \vartheta_p} c_j||W^P\mathbf{A}_j||_2\right)\right\}
\tag{14}
$$

where $\mathbf{L}$ is a graph Laplacian matrix constructed from $w_{nm}, n, m = 1, \ldots, N$ with $w_{nm} = e^{||\mathbf{y}_n - \mathbf{y}_m||^2/2B\eta^2}$ for neighboring pixels; otherwise, 0. $\mathrm{Tr}(\cdot)$ is the trace of the matrix, $P$ denotes the number of superpixels, $W^p$ are defined in Equation (9), and $c_j$ are defined in Equations (8), (10), and (13), with $\beta_1$ controlling smoothness and $\beta_2$ controlling sparsity of the abundance maps.

According to the method analysis of GMM in the Section 2,

$$
p(\mathbf{Y}|\mathbf{A}, \mathbf{\Theta}, \mathbf{D}) = \prod_{n=1}^{N} p(\mathbf{y}_n|\boldsymbol{\alpha}_n, \mathbf{\Theta}, \mathbf{D}).
\tag{15}
$$

Then, from the conditional density function $p(\mathbf{Y}|\mathbf{A}, \mathbf{\Theta}, \mathbf{D})$ and the abundance priors $p(\mathbf{A})$, taking Bayes' theorem, we can obtain the posterior:

$$
p(\mathbf{A}, \mathbf{\Theta}|\mathbf{Y}, \mathbf{D}) \propto p(\mathbf{Y}|\mathbf{A}, \mathbf{\Theta}, \mathbf{D})p(\mathbf{A})p(\mathbf{\Theta}),
\tag{16}
$$

where the GMM parameters $p(\mathbf{\Theta})(\mathbf{\Theta} := \pi_{jk}, \boldsymbol{\mu}_{jk}, \mathbf{\Sigma}_{jk} : j = 1, ..., M, k = 1, ..., K_j, \mathbf{m}_{nj} : j = 1, ..., M)$ is assumed to be an uniform distribution. As maximizing $p(\mathbf{A}, \mathbf{\Theta}|\mathbf{Y}, \mathbf{D})$ is equivalent to minimizing $-\log p(\mathbf{A}, \mathbf{\Theta}|\mathbf{Y}, \mathbf{D})$, combining Equations (4) and (13)–(15), we can obtain the objective function as

$$
\varepsilon(\mathbf{A}, \mathbf{\Theta}) = -\sum_{n=1}^{N} \log \sum_{\mathbf{k} \in \mathcal{K}} \pi_{\mathbf{k}} \mathcal{N}(\mathbf{y}_n|\boldsymbol{\mu}_{n\mathbf{k}}, \mathbf{\Sigma}_{n\mathbf{k}}) + \varepsilon_{\text{prior}}(\mathbf{A}),
$$
$$
\text{s.t. } \pi_{\mathbf{k}} \geq 0, \sum_{\mathbf{k} \in \mathcal{K}} \pi_{\mathbf{k}} = 1, \alpha_{nj} \geq 0, \sum_{j=1}^{M} \alpha_{nj} = 1, \forall n,
\tag{17}
$$

where $\varepsilon_{prior}(\mathbf{A}) = \sum_{p=1}^{P}(-\frac{\beta_1}{2}\mathrm{Tr}\left(\mathbf{A}^T\mathbf{L}\mathbf{A}\right) + \frac{\beta_2}{2}\sum_{\mathbf{A}_j \in \vartheta_p} c_j||W^P\mathbf{A}_j||_2)$ and $\boldsymbol{\mu}_{n\mathbf{k}}, \mathbf{\Sigma}_{n\mathbf{k}}$ are defined in Equation (5).

### 3.2. Optimization of the Proposed SGSGMM

Due to the flexibility of the EM algorithm, it can also be regarded as a kind of particular case of majorization-minimization algorithms [38], and we choose this algorithm to solve the SGSGMM model. Considering that the parameters in the abundance matrix $\mathbf{A}$ and $\Theta$ of the GMMs need to be iterated in each M step, here we use the generalized-expectation maximization (GEM) optimization algorithm to solve, as long as the complete data log-likelihood increases [32].

In each superpixel, the GEM algorithm is divided into E steps and M steps in the optimization solution. In the E step: we calculate the posterior probability of each implied variable $\gamma_{n\mathbf{k}}$ based on the observed data and initialization parameters. The mathematical form can be expressed as

$$\gamma_{n\mathbf{k}} = \frac{\pi_{\mathbf{k}} \mathcal{N}(\mathbf{y}_n | \boldsymbol{\mu}_{n\mathbf{k}}, \boldsymbol{\Sigma}_{n\mathbf{k}})}{\sum_{\mathbf{k} \in \mathcal{K}} \pi_{\mathbf{k}} \mathcal{N}(\mathbf{y}_n | \boldsymbol{\mu}_{n\mathbf{k}}, \boldsymbol{\Sigma}_{n\mathbf{k}})}. \tag{18}$$

In the M step, we will maximize the expected value of the log-likelihood probability. According to the Bayesian formula, incorporated the priors of $\mathbf{A}$, the final objective function $\varepsilon_M$ we need to minimize can be expressed as

$$\varepsilon_M = -\sum_{n=1}^{N} \sum_{\mathbf{k} \in \mathcal{K}} \gamma_{n\mathbf{k}} \left\{ \log \pi_{\mathbf{k}} + \log \mathcal{N}(\mathbf{y}_n | \boldsymbol{\mu}_{n\mathbf{k}}, \boldsymbol{\Sigma}_{n\mathbf{k}}) \right\} + \varepsilon_{prior}, \tag{19}$$

where the $\varepsilon_{prior}$ are defined in Equation (15). The weight of the Gaussian mixture $\pi_{\mathbf{k}}$ can be updated as

$$\pi_{\mathbf{k}} = \frac{1}{N} \sum_{n=1}^{N} \gamma_{n\mathbf{k}}. \tag{20}$$

In the next step, we need to focus on updating the $\boldsymbol{\mu}_{jk}, \boldsymbol{\Sigma}_{jk}$, and $\mathbf{A}$. Using Equation (5), we can obtain the derivatives of the objective function $\varepsilon_M$ in Equation (15) with respect to $\boldsymbol{\mu}_{jk}, \boldsymbol{\Sigma}_{jk}$, and $\alpha_{nj}$:

$$\frac{\partial \varepsilon_M}{\partial \boldsymbol{\mu}_{jl}} = -\sum_{n=1}^{N} \sum_{\mathbf{k} \in \mathcal{K}} \delta_{lk_j} \alpha_{nj} \boldsymbol{\lambda}_{n\mathbf{k}}, \tag{21}$$

$$\frac{\partial \varepsilon_M}{\partial \boldsymbol{\Sigma}_{jl}} = -\sum_{n=1}^{N} \sum_{\mathbf{k} \in \mathcal{K}} \delta_{lk_j} \alpha_{nj}^2 \boldsymbol{\Psi}_{n\mathbf{k}}, \tag{22}$$

$$\begin{aligned}
\frac{\partial \varepsilon_M}{\partial \alpha_{nj}} = &- \sum_{\mathbf{k} \in \mathcal{K}} \boldsymbol{\lambda}_{n\mathbf{k}}^T \boldsymbol{\mu}_{jk_j} - 2\alpha_{nj} \sum_{\mathbf{k} \in \mathcal{K}} \mathrm{Tr}(\boldsymbol{\Psi}_{n\mathbf{k}}^T \boldsymbol{\Sigma}_{jk_j}) \\
&+ \beta_1 (\mathbf{LA})_{nj} - \beta_2 \left( \frac{c_j}{||\mathbf{W}^p \mathbf{A}||_2} (\mathbf{W}^p)^T \mathbf{W}^p \mathbf{A} \right)_{nj},
\end{aligned} \tag{23}$$

where $\mathbf{W}^p$ is defined in Equation (9). $\mathbf{L}$ is the graph Laplacian matrix; $\delta_{lk_j} = 1$ when $l = k_j$, otherwise 0. $\boldsymbol{\lambda}_{n\mathbf{k}} \in \mathbb{R}^{B \times 1}$ and $\boldsymbol{\Psi}_{n\mathbf{k}} \in \mathbb{R}^{B \times B}$ are given by

$$\boldsymbol{\lambda}_{n\mathbf{k}} = \gamma_{n\mathbf{k}} \boldsymbol{\Sigma}_{n\mathbf{k}}^{-1} (\mathbf{y}_n - \boldsymbol{\mu}_{n\mathbf{k}}), \tag{24}$$

$$\boldsymbol{\Psi}_{n\mathbf{k}} = \frac{1}{2} \gamma_{n\mathbf{k}} \boldsymbol{\Sigma}_{n\mathbf{k}}^{-T} (\mathbf{y}_n - \boldsymbol{\mu}_{n\mathbf{k}}) (\mathbf{y}_n - \boldsymbol{\mu}_{n\mathbf{k}}^T) \boldsymbol{\Sigma}_{n\mathbf{k}}^{-T} - \frac{1}{2} \gamma_{n\mathbf{k}} \boldsymbol{\Sigma}_{n\mathbf{k}}^{-T}, \tag{25}$$

For the convenience of implementation, we can rewrite the derivatives Equations (23)–(25) in matrix forms as

$$\frac{\partial \varepsilon_M}{\partial \boldsymbol{\mu}_{jl}} = -\sum_{\mathbf{k} \in \mathcal{K}} \delta_{lk_j} (\mathbf{A}^T \boldsymbol{\Lambda}_{\mathbf{k}})_j, \tag{26}$$

$$\frac{\partial \varepsilon_M}{\partial \mathrm{vec}(\boldsymbol{\Sigma}_{jl})} = -\sum_{\mathbf{k} \in \mathcal{K}} \delta_{lk_j} ((\mathbf{A} \circ \mathbf{A})^T \boldsymbol{\Psi}_{\mathbf{k}})_j, \tag{27}$$

$$\frac{\partial \varepsilon_M}{\partial \mathbf{A}} = - \sum_{\mathbf{k} \in \mathcal{K}} \mathbf{\Lambda_k} \mathbf{R}_\mathbf{k}^T - 2\mathbf{A} \circ \sum_{\mathbf{k} \in \mathcal{K}} \mathbf{\Psi_k} \mathbf{S}_\mathbf{k}^T + \beta_1(\mathbf{LA}) - \beta_2((\mathbf{W}^p)^T \mathbf{W}^p \mathbf{A}^p \mathbf{B}^p), \tag{28}$$

where $\circ$ denotes the Hadamard product. $\mathbf{\Lambda_k} \in \mathbb{R}^{N \times B}$, $\mathbf{\Psi_k} \in \mathbb{R}^{N \times B^2}$ denote the matrices formed by $\lambda_{n\mathbf{k}}$, $\mathbf{\Psi}_{n\mathbf{k}}$, and $\mathbf{B}^p \in \mathbb{R}^{n_p \times n_p}$ is a diagonal matrix, which can be represented as

$$\mathbf{\Lambda_k} := [\lambda_{1\mathbf{k}}, \lambda_{2\mathbf{k}}, ..., \lambda_{N\mathbf{k}}]^T,$$

$$\mathbf{\Psi_k} := [\text{vec}(\mathbf{\Psi_{1k}}), \text{vec}(\mathbf{\Psi_{1k}}), ..., \text{vec}(\mathbf{\Psi_{Nk}})]^T,$$

$$\mathbf{B}^p = \text{diag}(\frac{c_1}{||\mathbf{W}^p \mathbf{A}_1||_2}, \frac{c_2}{||\mathbf{W}^p \mathbf{A}_2||_2}, ..., \frac{c_{np}}{||\mathbf{W}^p \mathbf{A}_{np}||_2})$$

$\text{vec}(\cdot)$ denotes the vectorization of the matrix and $\mathbf{R_k} \in \mathbb{R}^{M \times B}$, $\mathbf{S_k} \in \mathbb{R}^{M \times B^2}$ are defined by

$$\mathbf{R_k} := [\mu_{1k_1}, \mu_{2k_2}, ..., \mu_{Mk_M}]^T, \tag{29}$$

$$\mathbf{S_k} := [\text{vec}(\mathbf{\Sigma}_{1k_1}), \text{vec}(\mathbf{\Sigma}_{2k_2}), ..., \text{vec}(\mathbf{\Sigma}_{Mk_M})]^T. \tag{30}$$

Then, given an initial $\mathbf{A}$, we can update $\gamma_{n\mathbf{k}}$ and $\mathbf{A}$ alternately until convergence. In the choice of step size, we adopt the method in [18], decreasing Equations (26)–(28) by project gradient decent in each M step.

### 3.3. Model Selection

As can be seen from the analysis in the previous section, suppose we have a library of endmember spectra $\mathbf{Y}_j \in \mathbb{R}^{N_j \times B} : j = 1, ..., M$, with which we can estimate the GMM parameters $\mathbf{\Theta} := \pi_{jk}, \mu_{jk}, \mathbf{\Sigma}_{jk} : j = 1, ..., M$ by the standard EM algorithm. However, the number of the components $K_j$ for each endmember will be difficult to predict. If the number of components $K_j$ is set incorrectly, it will seriously affect the simulated distribution of the endmembers and greatly affect the unmixing accuracy of the model. To achieve adaptive selection of $K_j$, we use a model selection method based on cross-validation-based information criterion (CVIC) to adaptively select the number of components [39,40].

Suppose $\mathbf{Y}_j$ is the input pure pixels for the $j$th endmember, we will divide the input set $\mathbf{Y}_j$ into $V = 5$ subsets $(\mathbf{Y}_j^1, \mathbf{Y}_j^2, ..., \mathbf{Y}_j^5)$ with equal size. Then, for the each subset $\mathbf{Y}_j^v$, the remaining data in the subset are used to replace $\mathbf{Y}_j$ in turn and calculate the cross-validation evaluation criteria:

$$\mathcal{L}_{K_j} = \sum_v \mathcal{L}_{\mathbf{Y}_j^v}(\mathbf{\Theta}_j^v) \tag{31}$$

where $\mathcal{L}_{\mathbf{Y}_j^v}(\mathbf{\Theta}_j^v)$ is expressed as the distribution difference between the real distribution and the simulated distribution. Here, we choose the Kullbcal–Leibler (KL) divergence to measure the difference between probability distribution. The KL divergence can be mathematically written as

$$\mathcal{D}_{KL}(g_{m_j} || f_{m_j}) = \int_{\mathbb{R}^B} g_{m_j}(\mathbf{y}) \log \frac{g_{m_j}(\mathbf{y})}{f_{m_j}(\mathbf{y}|\mathbf{\Theta}_j)} d\mathbf{y}$$

$$\approx -\frac{1}{N_j} \sum_{n=1}^{N_j} \log f_{m_j}(\mathbf{y}_n^j | \mathbf{\Theta}_j) + \text{const}, \tag{32}$$

where $N_j$ denotes number of the $j$th endmember and $\mathbf{y}_n^j$ denotes $n$th element for the $j$th endmember. Then, maximizing $\mathcal{L}_{K_j}$ to find the corresponding $K_j$ as the final number of components $\hat{K} = \arg\max_{K_j} \mathcal{L}_{K_j}$.

*3.4. Implementation Issues*

In this section, we introduce the implementation details of the algorithm. For better comparison with other algorithms, the proposed method SGSGMM is tested in two scenarios: supervised and unsupervised. In the unsupervised unmixing scenario, the pixel purity assumption is assumed, which means there are enough pure pixels samples in the hyperspectral image for training to model the endmember probability distributions. As the optimization for GMM is nonconvex, the initialization will seriously affect the accuracy of the unmixing. Here, we use three different EE initialization methods to test the robustness of our algorithm: (1) K-means initialization, (2) Vertex Component Analysis (VCA) initialization [41], and (3) region-based VCA initialization [42].

Taking the endmember initialization to find the initial $\mathbf{R}_1$, we start with $K_j = 1$, and the initial abundance $\mathbf{A}$ is set to $\mathbf{A} \leftarrow \mathbf{Y}\mathbf{R}_1^T \left( \mathbf{R}_1 \mathbf{R}_1^T + \epsilon \mathbf{I}_M \right)^{-1}$ (by minimizing $||\mathbf{Y} - \mathbf{A}\mathbf{R}_1||_F^2$). The covariance matrices and noise matrices are set to $\mathbf{\Sigma}_{j1} = 0.1 I_B$ and $\mathbf{D} = 0.001^2 I_B$, respectively. Then, we adopt the GEM algorithm to iteratively update under the initial conditions. When the number of iterations reaches the predefined value, we can obtain relatively pure pixel $\mathbf{Y}_j$ by thresholding the abundance (e.g., $\alpha_{nj} > 0.99$); then, adapting the method in Section 3.3 and taking $\mathbf{Y}_j$ as input to estimate the number of components $K_j$, continue iterating $\mathbf{\Theta}$ and $\mathbf{A}$ to algorithm convergence.

For the supervised unmixing scenario, the difference between the above unsupervised method is that the library of the endmember spectra is assumed known. Thus, we could directly take the pure pixel $\mathbf{Y}_j$ as input, adopt CVIC to estimate the number of components $K_j$ and using GEM to iterate. Therefore no need to set the initial conditions of the algorithm. The detailed procedure of the supervised and unsupervised cases are listed in Algorithms 1 and 2, respectively.

In the specific experiments, the number of endmembers is known in advance, and it can also be automatically detected by the HySime method [43]. In both cases, PCA is adapted to reduce the high computational cost. All thresholds involved in this paper are the convergence value of the algorithm and the other is the CVIC threshold; the convergence threshold is set as 0.2% and the CVIC threshold is set as 1%. For fairness of comparison, the maximum number of iterations for all the algorithms is set to 100. The weighting spectral and spatial terms $\lambda$ is set to $\lambda = 0.5$.

---

**Algorithm 1** Details for unsupervised of SGSGMM

---

**Input:** Collected mixed pixel matrix $\mathbf{Y}$; the parameter of smoothness and spatial sparsity constraint

　　$\beta_1, \beta_2$, and the weight value for the distance metric $\lambda$;

**Output:** The estimated abundance matrix $\mathbf{A}$;

  1: **Preprocessing:**
  2:　　(a) Implement PCA and generate **P** spatial groups based on SLIC
  3:　　(b) Take initialization to find initial $\mathbf{R}_1$ and set $\mathbf{A} \leftarrow \mathbf{Y}\mathbf{R}_1^T \left( \mathbf{R}_1 \mathbf{R}_1^T + \epsilon \mathbf{I}_M \right)^{-1}$
  4:　　(c) Calculate weight matrix $\mathbf{W}^p$ by Equation (9), the confidence index $c_j$ by Equations (8) and (10)
  5: **for each superpixel:**
  6: **while** not converged **do**
  7:　　E step: Calculate $\gamma_{n\mathbf{k}}$ by Equation (19)
  8:　　M step: Calculate derivatives of $\mu_{jk}, \mathbf{\Sigma}_{jk}, \alpha_{nj}$ by Equations (23)–(25)
  9:　　Update $\gamma_{n\mathbf{k}}, \mathbf{A}, \mu_{jk}, \mathbf{\Sigma}_{jk}$ and $\mathbf{W}^p$.
 10:　　**if** iterations > predefined iterations
 11:　　　Set pure pixels $\mathbf{Y}_j = \mathbf{Y}((\mathbf{A}(:,j) == 1), :)$, and estimate $K_j$ by Equations (31) and (33).
 12:　　　Go to E step
 13:　　**end if**
 14: **end while**

---

---

**Algorithm 2** Details for supervised of SGSGMM

---

**Input:** Collected mixed pixel matrix **Y**, endmember **E**; the parameter of smoothness and spatial

　　　sparsity constraint $\beta_1, \beta_2$, and the weight value for the distance metric $\lambda$;

**Output:** The estimated abundance matrix **A**;

　　**Preprocessing:**

2:　　(a) Implement PCA and Generate **P** spatial groups based on SLIC

　　　(b) the confidence index $c_j$ by Equations (8) and (10)

4: Take endmember **E** as input, using CVIC to estimate $K_j$ and calculate $\mu_{jk}, \Sigma_{jk}$ by standard EM

　　Set $\mathbf{A} \leftarrow \mathbf{Y}\mathbf{R}_K^T \left(\mathbf{R}_K\mathbf{R}_K^T + \epsilon\mathbf{I}_M\right)^{-1}$ as initialization

6: **for each superpixel:**

　　**while** not converged **do**

8:　　E step: Calculate $\gamma_{n\mathbf{k}}$ by Equation (19)

　　　M step: Calculate derivatives of $\mu_{jk}, \Sigma_{jk}, \alpha_{nj}$ by Equations (23)–(25)

10:　　Update $\gamma_{n\mathbf{k}}, \mathbf{A}, \mu_{jk}, \Sigma_{jk}$ and $\mathbf{W}^p$.

　　**end while**

---

## 4. Experimental Results

In this section, we will compare the proposed SGSGMM with some state-of-the-art unmixing methods—NCM [20], BCM (spectral version with quadratic programming) [21], GMM [22], and SGSNMF [31]—in both synthetic and real datasets. BCM and NCM are both supervised algorithms. Thus, we implement those methods with the pure pixels taken as input and results are the abundance maps. For the GMM and SGSGMM algorithms, as the number of components $K_j$ will affect the calculation rate of the algorithm, to accelerate the computation time of iteration, the original data is reduced to 10 dimensions by PCA as input.

In the quantitative comparison of abundance, we calculate the root-mean-squared error (RMSE) for abundance error, which is defined as

$$\text{RMSE} = \left(\frac{1}{N}\sum_n |\alpha_{nj}^{GT} - \alpha_{nj}^{est}|^2\right)^{1/2}, \tag{33}$$

where $\alpha_{nj}^{GT}$ denotes the ground truth and $\alpha_{nj}^{est}$ denotes the corresponding estimated abundances. For the real HSI dataset, the RMSE of abundance error is calculated by $error_j = \left(\frac{1}{|\mathcal{I}|}\sum_{n\in\mathcal{I}} |\alpha_{nj}^{GT} - \alpha_{nj}^{est}|^2\right)^{1/2}$, as only some pure pixels in the real dataset are recognized as ground truth; here, the $\mathcal{I}$ denotes the pure pixel index set.

### 4.1. Synthetic Datasets

To verify the ability in estimation of both endmembers and fractional abundances, and the robustness of our algorithm under different initialization conditions, for the synthetic dataset, our algorithm is tested in the unsupervised case. To better simulate the endmember variability existing in the real hyperspectral images, the spectral of the endmember seeds are randomly selected from the ASTER spectral library [18]: limestone, basalt, concrete, conifer, and asphalt (Figure 2). The covariance of the endmember spectra are based on those endmember seeds with slight constant variations. The endmember spectra range is including three parts, which is visible and near-infrared (VNIR; 0.4 μm to 1.0 μm), the short-wavelength infrared (SWIR; 1.0 μm to 2.4 μm), and the thermal infrared (TIR; 8 μm to 12 μm). Their covariance matrices are constructed by $\alpha_{jk}^2\mathbf{I}_B + b_{jk}^2\mu_{jk}\mu_{jk}^T$, where $\mu_{jk}$ is a unit vector controlling the major variation direction. The endmember spectral we used to generate

the synthetic data are shown in Figure 3, where we can intuitively see the centers and variations of the endmember spectral signatures.

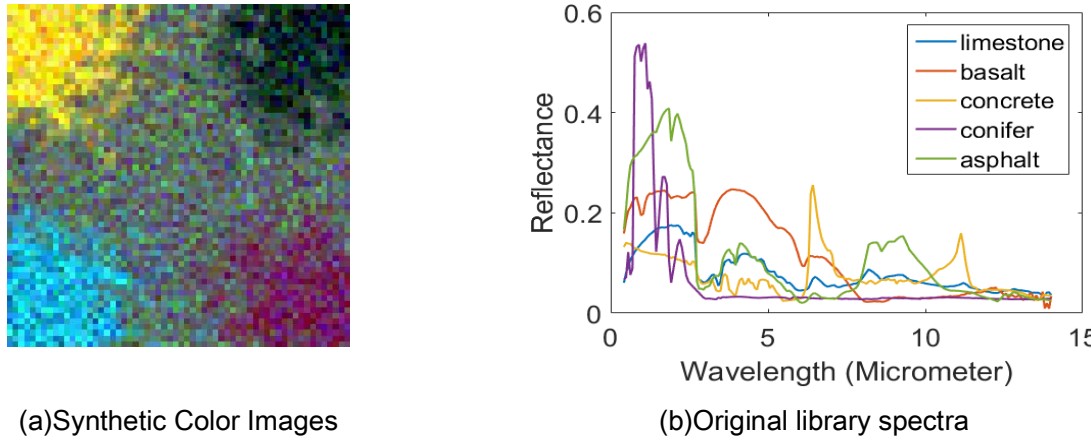

(a)Synthetic Color Images                    (b)Original library spectra

**Figure 2.** (**a**) The color images of the synthetic dataset. (**b**) The spectral of the endmember seeds used to construct the synthetic dataset, which is all extracted from the ASTER library.

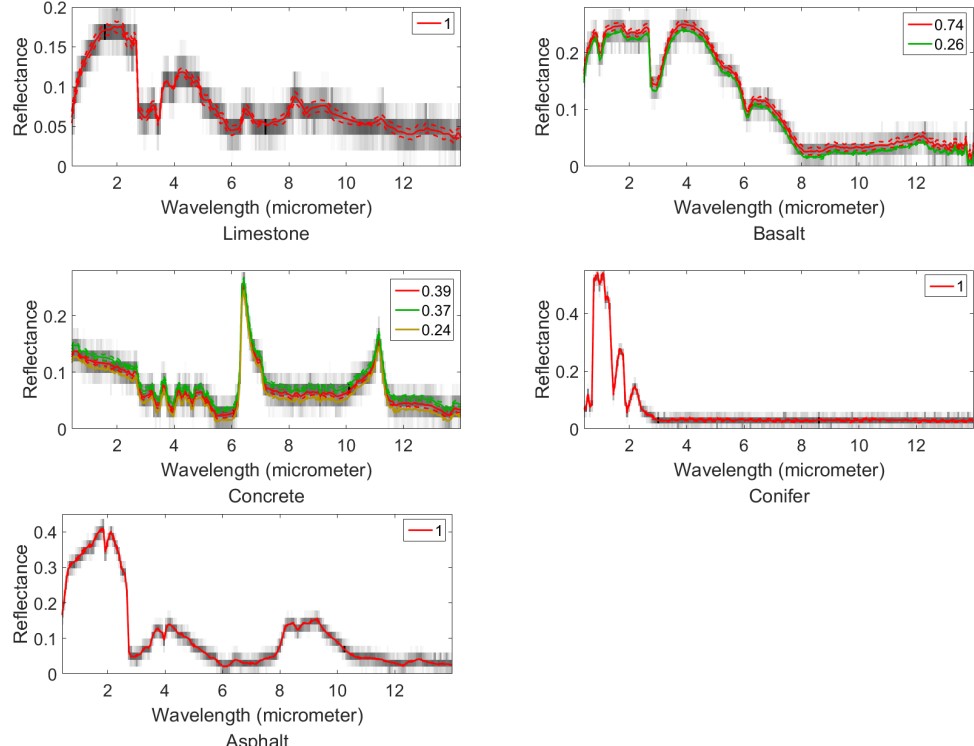

**Figure 3.** The endmember spectral signatures of the synthetic dataset. The gray portion of the background within the image represents the reflection value of the pure pixel reflectance at each wavelength position. The different colors in the figure represent different components, and the corresponding legend indicates its prior probability. The solid line indicates the center of Gaussians $\mu_{jk}$, and the dotted line indicates the variance range of each Gaussian component, which is constructed by $\mu_{jk} \pm 2\sqrt{\hat{\sigma}_{jk}\mathbf{v}_{jk}}$ ($\hat{\sigma}_{jk}$ denotes the largest eigenvalue of $\Sigma_{jk}$; $\mathbf{v}_{jk}$ denotes the corresponding eigenvector).

The size of the synthetic dataset we constructed is $60 \times 60$. We choose one material as background, the other materials are randomly placed in the corner, whose shape, width, and location are both sampled from Gaussian distributions. Also, to allow the pixels to have a random value, the abundances

are sampled from the Dirichlet distribution. The specific generation step of the abundance map follows [44]. Here, the additive noises we added to the mixed pixels are assumed to follow the Gaussian distribution $\mathcal{N}(\mathbf{n}_n | 0, \mathbf{D})$ ($\mathbf{D} = \text{diag}\left(\sigma_1^2, \sigma_2^2, ..., \sigma_B^2\right) \in \mathbb{R}^{B \times B}$). Figure 2a shows the resulting color images by extracting the bands corresponding to wavelengths 488 nm, 556 nm, and 693 nm. The parameters in this experiment are superpixel size $w = 7$, the weighting spectral and spatial terms are $\lambda = 0.5$, $\beta_1 = 0.2$, and $\beta_2 = 0.1$.

(1) Accuracy of abundance maps comparison: Figure 4 shows the abundance maps comparison in the synthetic dataset. As NCM and BCM are both supervised methods, thus we take the endmember spectra library as input. For SGSGMM, GMM, and SGSNMF methods, the endmember are both taken region-based VCA methods as initialization [42]. Because the materials except the background are randomly placed in the four corners when the image is generated, the four endmembers' abundance map (basalt, limestone, conifer, and concrete) should look relatively clean and less cluttered. Comparing the ground truth (the first row of the Figure 4), we can see that although the size and shape of the GMM abundance maps are relatively consistent, some discrete and spatially isolated points are not well estimated. These points that should have abundance values are predicted as 0, which can be seen relatively clearly from the abundance map of asphalt and limestone. This is because the GMM uses the pixels of the entire image as the training when processing the image, and some discrete points are easily averaged out when the sparse priors constraints are performed in the entire image. It is further explained that the use of SS and group sparse constraints as a priori can better improve the unmixing accuracy within the GMM framework. For the BCM and NCM algorithms, although the discrete points are estimated normally, the shape of the abundance map is much different from the ground truth, which means that many pixels with an original abundance of 1 are predicted to have an abundance of 0. This is related to the fact that the probability distributions of the BCM and NCM models are not very close to the true endmember distribution. Although they all use the endmember spectra library as input, the performance is farther than the ground truth. For the SGSNMF, which also uses SS and group sparse constraints as the abundance priori, although it takes full use of the local spatial and spectral information within the hyperspectral image pixels, it does not consider the effects caused by endmember variability to unmixing. When the endmember spectra set is not fixed, the spectrum of the entire image may not be the same. The inaccuracy of the endmember signature will affect the performance of the unmixing to a great extent. Thus, the SGSNMF abundance maps also perform relatively poorly. Compared with these four algorithms, the result in SGSGMM is much closer to the ground truth map, which also shows the effectiveness of our proposed algorithm. The quantitative performance of the abundance map is shown in Table 1.

(2) Histograms of pure pixels comparison: The histograms of the pure pixels for the five materials are shown in Figure 5. As the BCM algorithm is not modeled as Gaussian distribution and SGSNMF is also not a distribution-based method, the histograms of the statistical probability value for the five materials is only compared among SGSGMM, GMM, and NCM algorithms. The estimated of each distribution is calculated by multiplying the density function value of each bin position by the bin size when projected to 1-dimensional space determined by performing PCA. The histograms in Figure 5 are the pure pixels for each material when projected to 1-dimensional space. The probability value of the histogram is the frequency statistics of pure pixels falling into different intervals after PCA dimensionality reduction. From Figure 5, we find that when the distribution of the pure materials is generated by an unimodal Gaussian, all the estimated distribution are similar, such as limestone, conifer, and asphalt. However, for basalt and concrete, where the probability distribution is the multipeak, SGSGMM and GMM can provide a more accurate estimation, because NCM assumes the endmembers for each pixel are sampled from single Gaussian. When comparing the two algorithms GMM and SGSGMM, most of their histograms are similar and fitted to the ground truth. Nevertheless, for basalt. SGSGMM provides a better estimation. This also confirms the point mentioned above that taking SS will help GMM to better separate the clusters, and reduce the risk of failing to estimated the distribution of the cluster. The quantitative analysis of these three algorithms is shown in Table 2,

we calculate the probability value in each histogram between the ground truth and estimate value by using RMSE Equation (33).

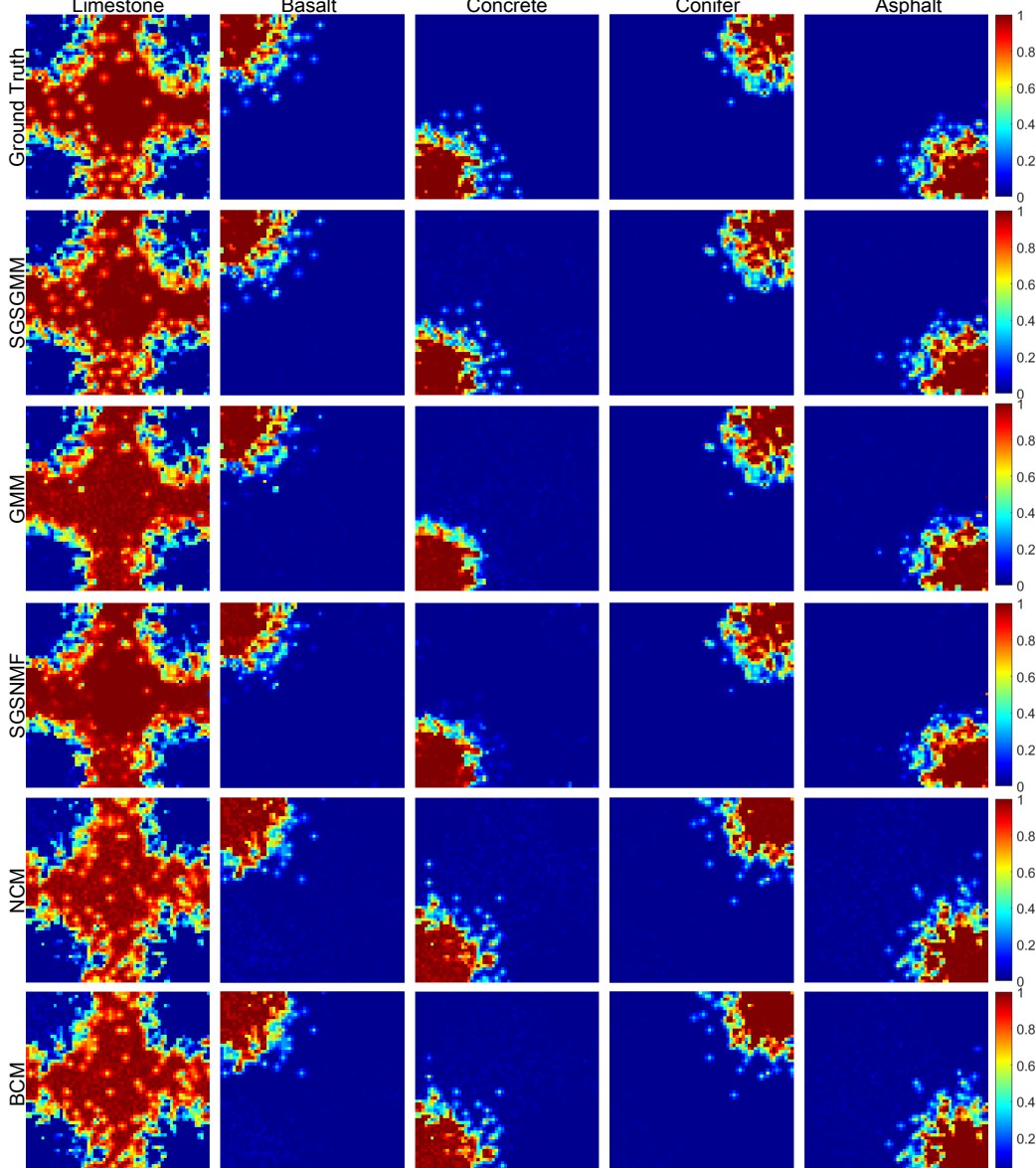

**Figure 4.** Abundance maps comparison for the ground truth, SGSGMM, Gaussian mixture model (GMM), spatial group sparsity regularized non-negative matrix factorization (SGSNMF), normal compositional model (NCM), and Beta compositional model (BCM).

**Table 1.** Abundance errors for synthetic dataset.

| $\times 10^{-3}$ | SGSGMM | GMM | SGSNMF | NCM | BCM |
|---|---|---|---|---|---|
| Asphalt | **208** | 459 | 672 | 566 | 743 |
| Shadow | **80** | 197 | 261 | 278 | 311 |
| Roof | **118** | 340 | 463 | 460 | 586 |
| Grass | **51** | 129 | 175 | 248 | 273 |
| Tree | **81** | 161 | 236 | 262 | 277 |
| Whole map | **107** | 257 | 359 | 363 | 438 |

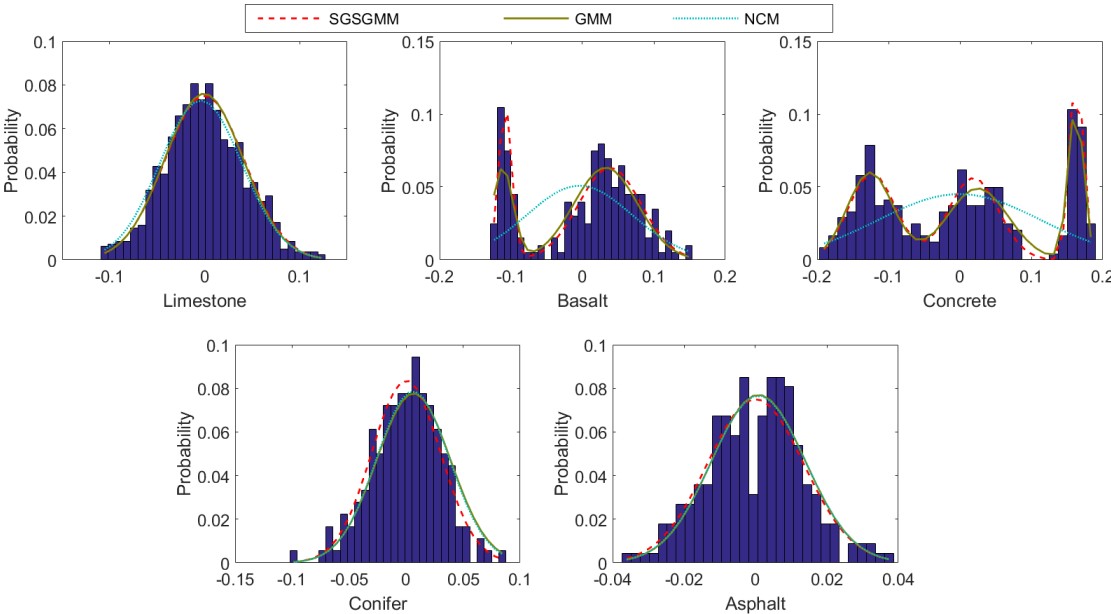

**Figure 5.** The histograms of the pure pixels for the 5 materials. The x-axis is expressed as a pure pixel for each material via PCA to 1-dimensional space, and the y-axis represents the proportion of occupying each bin size in the histogram. The probability of each distribution is calculated by multiplying the value of the density function at each bin location with the bin size.

**Table 2.** The root-mean-square error (RMSE) calculated between the probability value in each histogram and the estimated value at each bin location for the synthetic dataset.

| $\times 10^{-4}$ | SGSGMM | GMM | NCM |
|---|---|---|---|
| Limestone | 49 | **48** | 51 |
| Basalt | **114** | 138 | 285 |
| Concrete | **90** | 91 | 285 |
| Conifer | **66** | 85 | 81 |
| Asphalt | **115** | **115** | **115** |
| Mean | **87** | 95 | 163 |

### 4.2. Real-Data Experiments

Two real HSIs are also used to evaluate the unmixing accuracy. In these experiments, for the fairness of the experiment, all the algorithms are implemented as a supervised case and taking endmember spectral library as input.

#### 4.2.1. Mississippi Gulfport Datasets

For the real data experiments, the first real HSI was collected over the campus of Southern Mississippi–Gulfpark. The size of the Gulfport dataset is $271 \times 284$, and the spectra range is from 0.368 μm to 1.043 μm with 1 m/pixel spatial resolution. To better compare the unmixing results between the proposed method and GMM, the ROI we selected is a $120 \times 80$ area, which contains five materials: road, shadow, building, grass and tree. Compared with the previous ROI in [22], it contains more trees and asphalt materials and is larger in size. The selected ROI area and the corresponding abundance map are shown in Figure 6c,d. The superpixel map we used is shown in Figure 6e. The parameters in this experiment are superpixel size $w = 7$; the weighting spectral and spatial terms are $\lambda = 0.5$, $\beta_1 = 1$, and $\beta_2 = 0.1$.

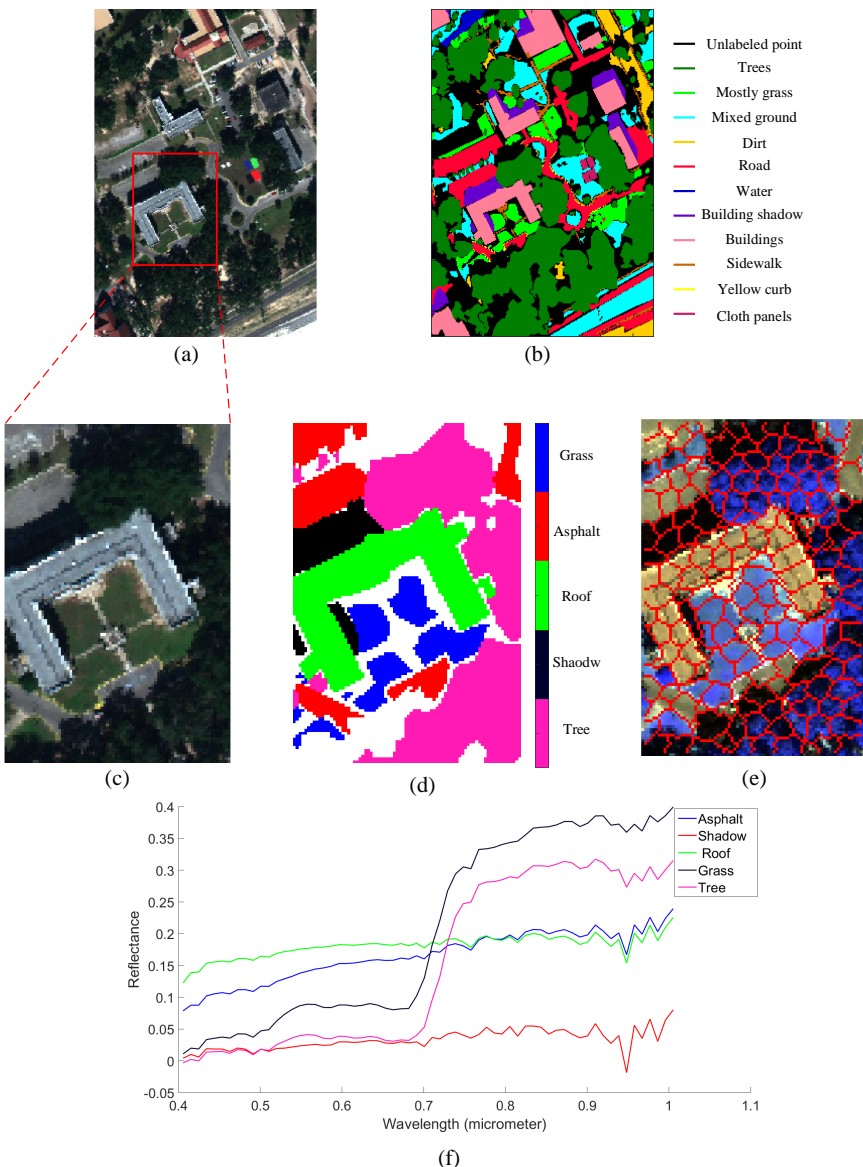

**Figure 6.** (**a**) The original RGB image. (**b**) The corresponding ground truth materials of Gulfport dataset. (**c**) The selected ROI area. (**d**) The corresponding ground truth materials in the ROI. (**e**) The superpixel map we used for the experiment. (**f**) The wavelength reflectance of mean spectra signature for the 5 materials.

Figure 7 shows the abundance maps comparison in the Gulfport dataset. We can see that SGSGMM matches the ground truth (the first row of Figure 7) best, followed by GMM. For NCM, BCM, and SGSNMF, we do not use PCA to get the main information while using the whole bands HSI dataset as input. Nevertheless, they could not provide a more accurate estimation. For example, the first and fourth abundance maps of NCM and BCM show that the pixels of asphalt and shadow are mixed with roof, and NCM fails to estimate the abundance of the tree area. For SGSNMF, although the shape and size of its abundance maps look good in general, some pure regions are inconsistent, which also shows the insufficiency in this case. The abundance maps of SGSGMM not only show sparse areas, but also explain adjacent boundaries. More specifically, in the comparison of the fifth abundance maps, SGSGMM provides a relatively accurate estimation, whereas other algorithms perform poorly. This is because there is a large number of homogeneous regions in the fifth material, therefore sparse

structures of the spatial groups can be more effectively exploited. The quantitative abundance errors of these algorithms are shown in Table 3, which also implies that SGSGMM performs best overall.

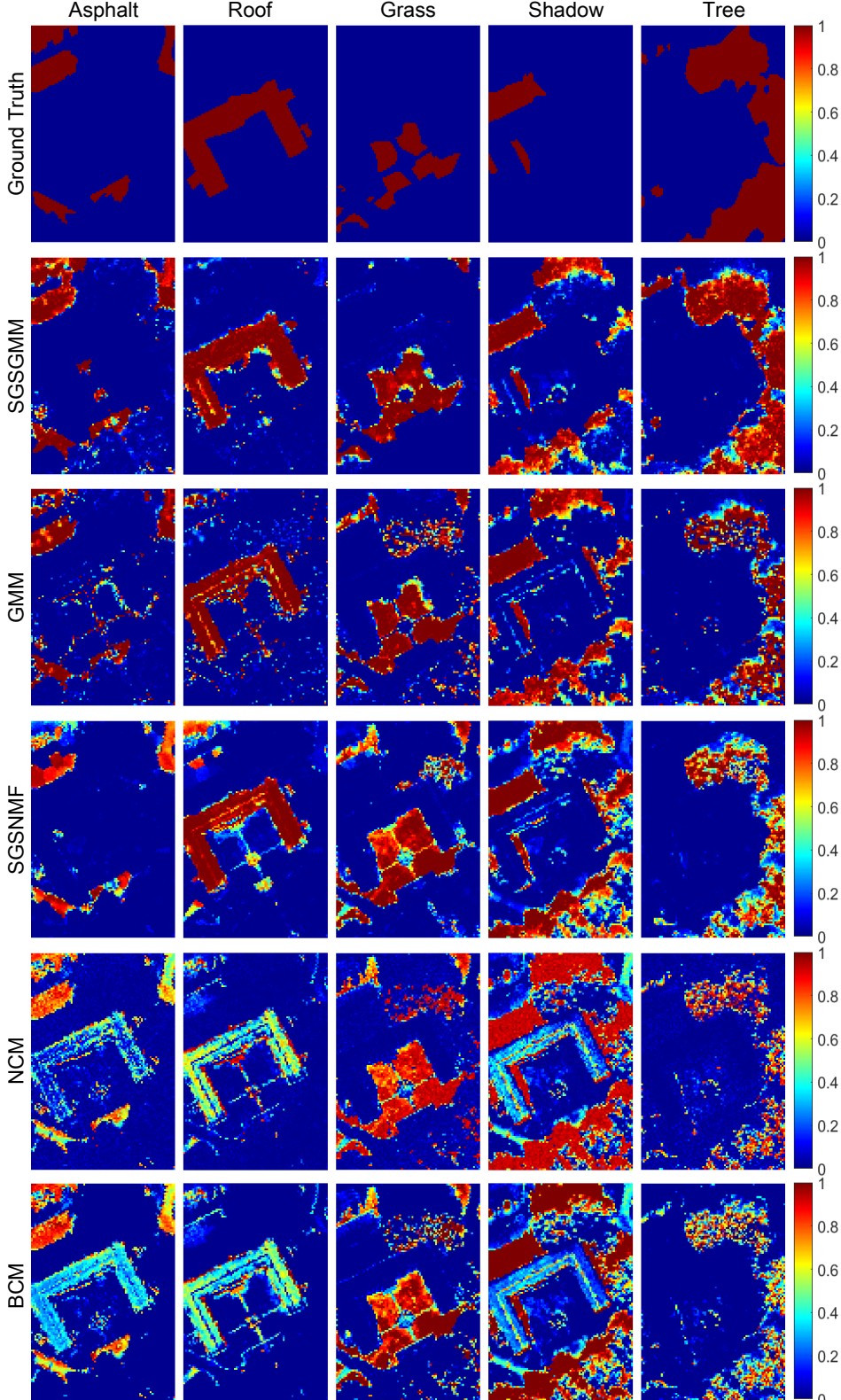

**Figure 7.** Abundance maps comparison for the ground truth, SGSGMM, GMM, SGSNMF, NCM, and BCM.

**Table 3.** Abundance errors for Gulfport dataset.

| $\times 10^{-3}$ | SGSGMM | GMM | SGSNMF | NCM | BCM |
|---|---|---|---|---|---|
| Asphalt | **189** | 384 | 513 | 474 | 440 |
| Roof | **220** | 333 | 286 | 647 | 660 |
| Grass | **57** | 67 | 95 | 183 | 130 |
| Shadow | 163 | 154 | 158 | 137 | **110** |
| Tree | **385** | 628 | 636 | 767 | 728 |
| Whole map | **158** | 276 | 280 | 357 | 351 |

The wavelength reflectance of endmember spectra for the Gulfport is shown in Figure 8. Figure 9 shows the histograms of the pure pixels for the five materials. The pure pixel for each material is determined via PCA to 1-dimensional space. Although most of the histograms are single peaks, NCM still performs poorly when estimating the endmember distribution. In contrast, our method and GMM algorithm are more suitable for this pure pixel distribution. Comparing the performance of all five materials, our algorithm can provide a better fit to the ground truth. The quantitative analysis presents in Table 4, which can also be verified. Noted that evaluation metric we use are slightly different from those used in [22], the specific details can be referred to the uploaded code.

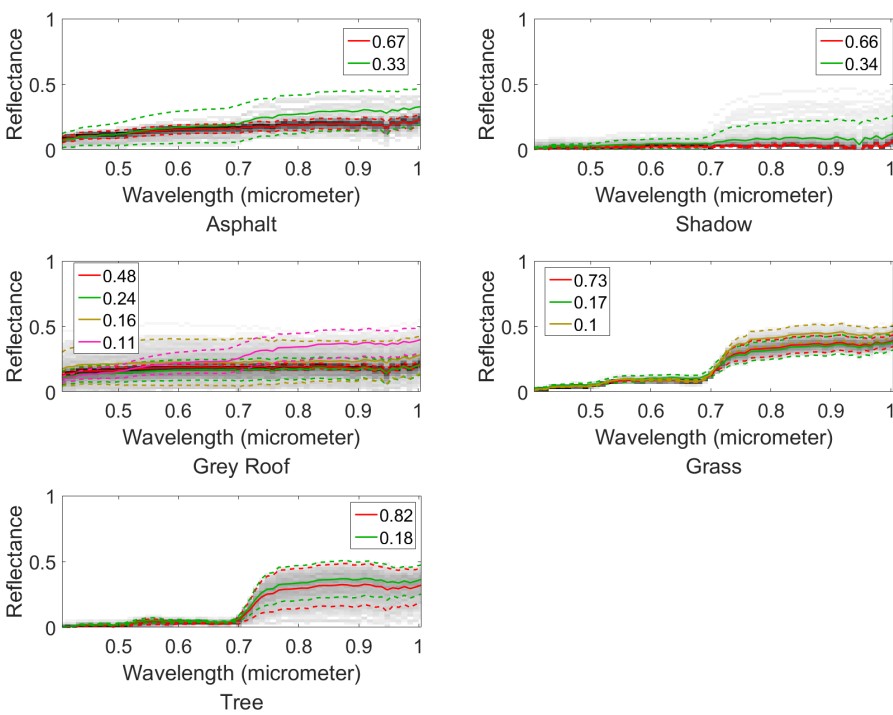

**Figure 8.** The wavelength reflectance space of the endmember signature estimated for the Gulfport dataset, which has the same meaning as in Figure 5.

**Table 4.** The RMSE calculated between the probability value in each histogram and the estimated value at each bin location for the Gulfport dataset.

| $\times 10^{-4}$ | SGSGMM | GMM | NCM |
|---|---|---|---|
| Asphalt | **100** | 232 | 178 |
| Roof | **72** | 227 | 305 |
| Grass | **41** | 71 | 134 |
| Shaodw | **219** | **219** | 566 |
| Tree | **54** | 90 | 112 |
| Mean | **97** | 168 | 259 |

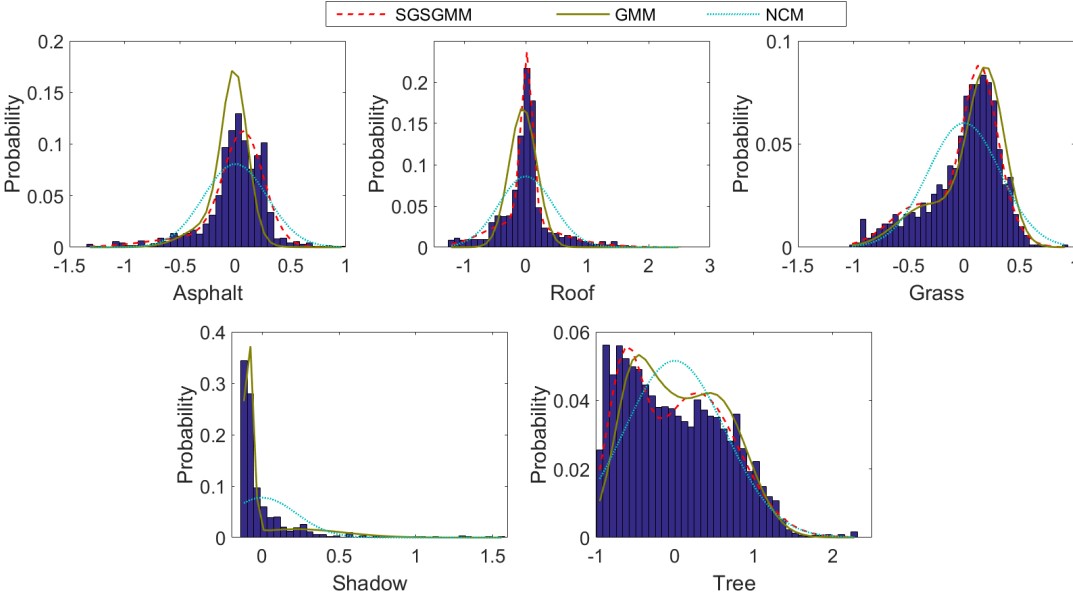

**Figure 9.** The estimated distributions and the histograms of pure materials for the SGSGMM, GMM, and NCM.

### 4.2.2. Salinas-A Datasets

The second real HSI is collected by the AVIRIS sensor over Salinas Valley, California, which is a $512 \times 217$ image with 224 bands and high spatial resolutions (3.7-m pixels). The ROI we choose to experiment with is a small sub-scenario Salinas image, denoted as Salinas-A. This is also commonly used. It contains $86 \times 83$ pixels, including six materials. The RGB image and corresponding abundance map are shown in Figure 10a,b. The superpixel map we used is shown in Figure 10c. The parameters in this experiment are set to: superpixel size $w = 7$, the weighting spectral and spatial terms $\lambda = 0.5$, $\beta_1 = 1$, and $\beta_2 = 0.1$.

The abundance maps comparison from different algorithms is shown in Figure 11. We can clearly see the inefficiency of NCM, BCM, and SGSGMM on this dataset. NCM, BCM, and SGSNMF all fail to estimate the pure pixels of corn, and SGSNMF performs too many inconsistent regions, which should be the pure material areas. SGSGMM matches the ground truth best, followed by GMM. The quantitative abundance errors of these algorithms are shown in Table 5, which also implies that SGSGMM performs best overall. Figure 12 shows the wavelength reflectance space of the endmember signature for the Salinas-A dataset. The estimated distributions and the histograms of pure pixels for the SGSGMM, GMM, and NCM are shown in Figure 13. We can see that for lettuce 7wk, NCM still does not fit the histograms, although the distribution is not multiple peaks. For lettuce 5wk, the GMM algorithm do not closely approximate the ground truth since the pure pixels are not single peaks. This also shows that our method can help the GMM better separate the clusters and enhance the performance in the estimated distribution. The quantitative analysis is presented in Table 6.

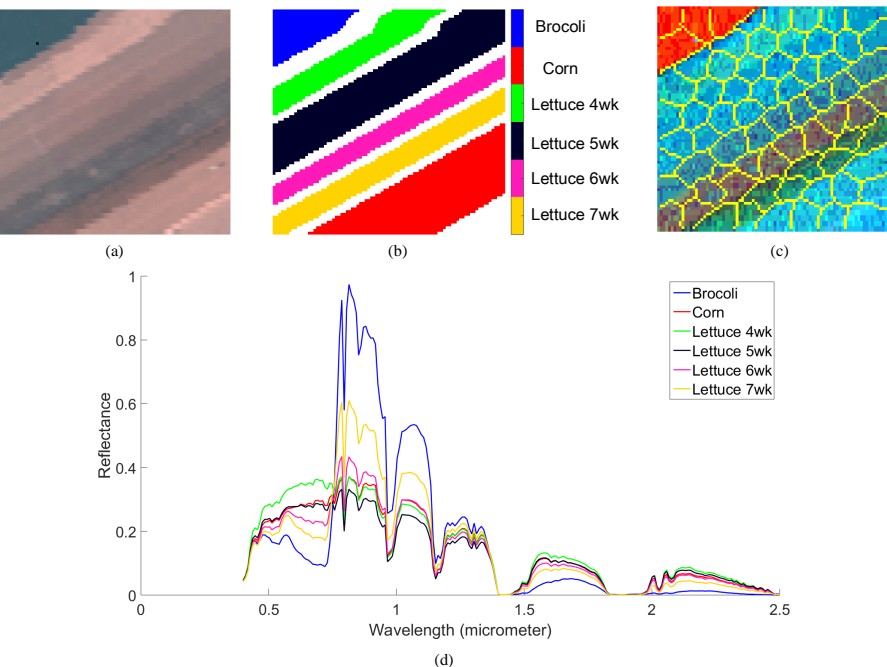

**Figure 10.** (**a**) The original RGB image. (**b**) The corresponding ground truth materials of Salinas-A dataset. (**c**) The superpixel map we used for the experiment. (**d**) The mean spectra of the 6 materials.

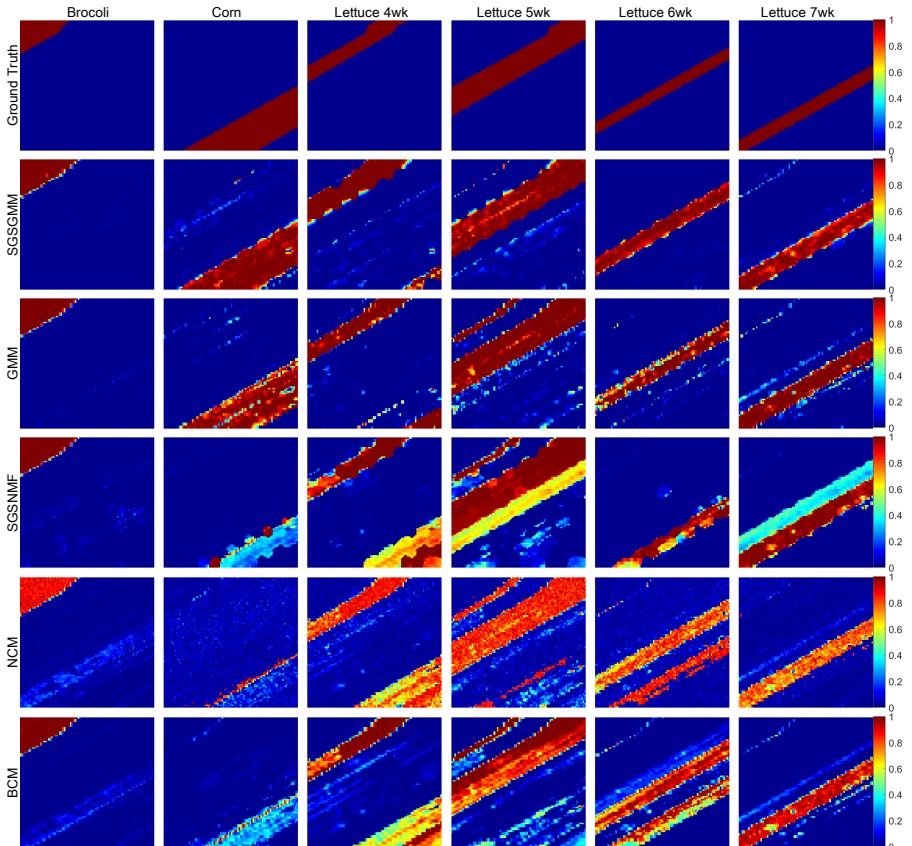

**Figure 11.** Abundance maps comparison for the ground truth, SGSGMM, GMM, SGSNMF, NCM, and BCM.

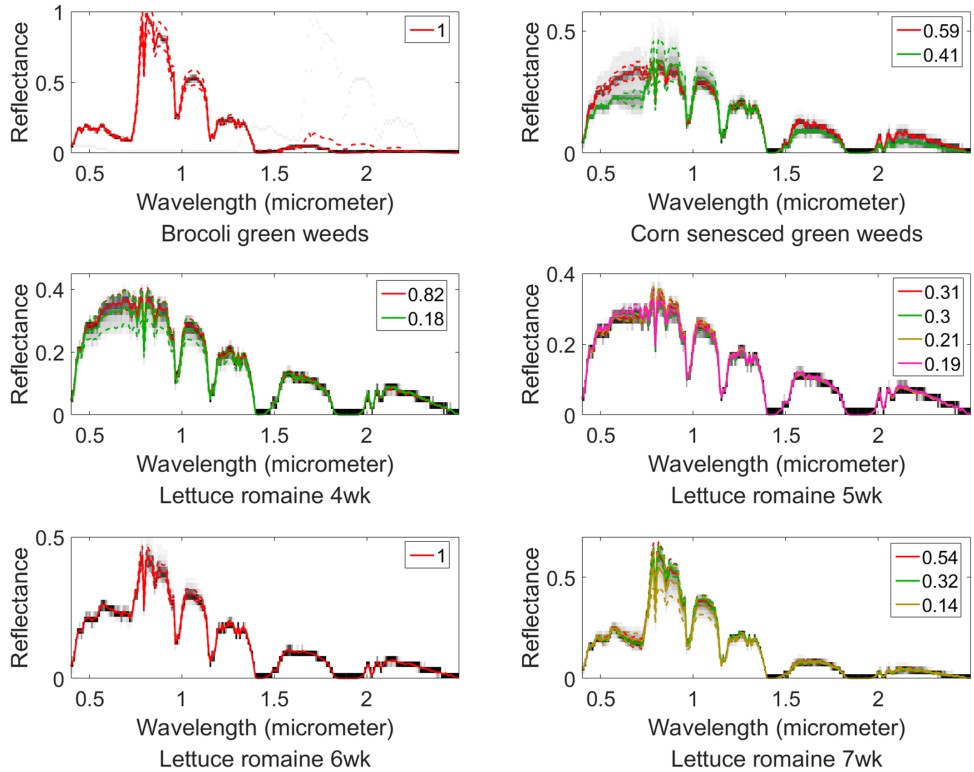

**Figure 12.** The wavelength reflectance space of the endmember signature estimated for the Salinas-A dataset, which has the same meaning as in Figure 5.

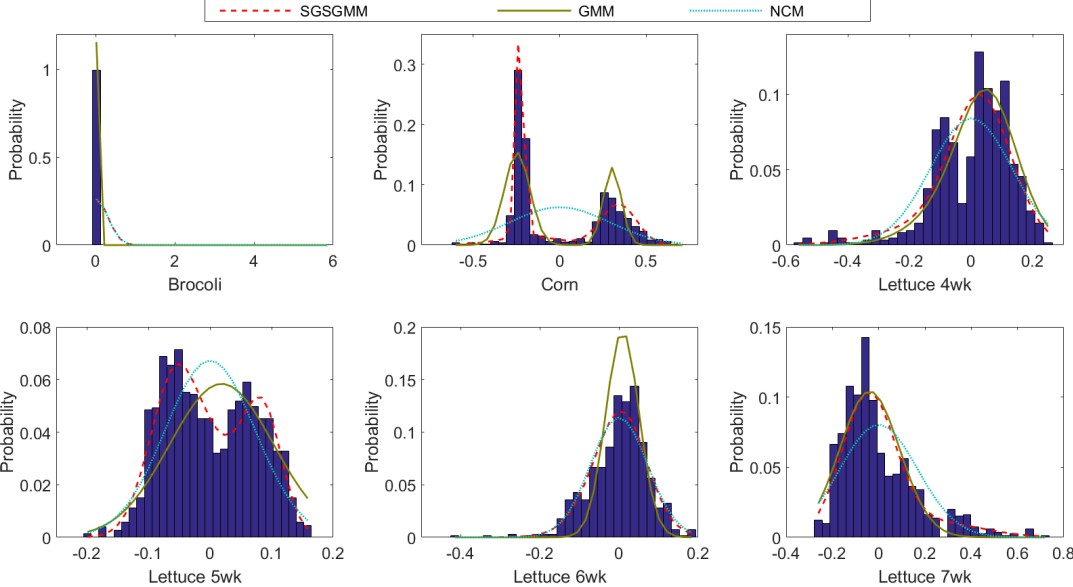

**Figure 13.** The estimated distributions and the histograms of pure materials for the SGSGMM, GMM, and NCM.

**Table 5.** Abundance errors for Salinas-A dataset.

| $\times 10^{-3}$ | SGSGMM | GMM | SGSNMF | NCM | BCM |
|---|---|---|---|---|---|
| Brocoli | 528 | 715 | **511** | 1421 | **511** |
| Corn | **1291** | 2087 | 8068 | 8790 | 8021 |
| Lettuce 4wk | **150** | 2096 | 2766 | 2732 | 2396 |
| Lettuce 5wk | 556 | 520 | **324** | 1858 | 1536 |
| Lettuce 6wk | **530** | 1975 | 9985 | 2529 | 1597 |
| Lettuce 7wk | **790** | 1046 | 1427 | 3053 | 2423 |
| Whole map | **407** | 802 | 2502 | 2268 | 2006 |

**Table 6.** The RMSE calculated between the probability value in each histogram and the estimated value at each bin location for the Salinas-A dataset.

| $\times 10^{-4}$ | SGSGMM | GMM | NCM |
|---|---|---|---|
| Brocoli | 315 | 291 | 317 |
| Corn | **140** | 382 | 586 |
| Lettuce 4wk | 177 | **172** | 196 |
| Lettuce 5wk | **63** | 150 | 150 |
| Lettuce 6wk | **116** | 233 | 134 |
| Lettuce 7wk | **151** | 163 | 215 |
| Mean | **160** | 231 | 266 |

## 5. Discussion

In this section, we will present an analysis of the sensitivity and efficiency of the algorithm and further discuss the limitations of the method.

(1) Sensitivity analysis to different initializations: As the endmember spectra library is not used as input in the unsupervised scenario, the initialization conditions will affect the accuracy of the unmixing to a large extent. In order to test the sensitivity and robustness of the proposed method. Here, we test the comparison algorithms under three different initialization conditions to compare the unmixing performance, including K-means, VCA, and region-based VCA. More specifically, K-means and VCA are looking for their candidate endmember in the original spectra, whereas region-based VCA searches for the candidate endmember in the average spectrum of each superpixel. The scatter plot of the endmember center under different initial conditions is shown in the Figure 14. We can see that the region-based initialization provide a more approximate and robust initial value relative to the optimal global solution compared with other initializations. This is because the region-based initial condition searches for candidate endmembers in each homogeneous region, and the original spatial information in the image can be well preserved under the correct superpixel groups. The specific abundance error values under different initial conditions are shown in the Table 7; we can see that the corresponding unmixing results within region-based method preform the highest accuracy, whereas the K-means initialization based performs poorly because it failed to estimate the center of the endmember well. Furthermore, in the comparison of the unmixing results within different algorithms, we can find that SGSGMM performs the best unmixing precision under all initialization conditions followed by GMM. The SGSNMF algorithm does not perform good unmixing accuracy even under the region-based initialization condition, although the local spatial information in the image is fully considered, which also illustrates the importance of considering the endmember variation problem to the accuracy of unmixing. For NCM and BCM, as they are both supervised methods, the algorithms are only implemented with the endmember library as input. Therefore, the experiments in different initialization situations are not considered. Comparing the results with these GMM, NCM, and BCM methods, which simulate endmember variability through probability distributions, we can also confirm that taking SS and group sparsity constraints can better capture the spatial data structure and enhance the performance in the abundance estimation.

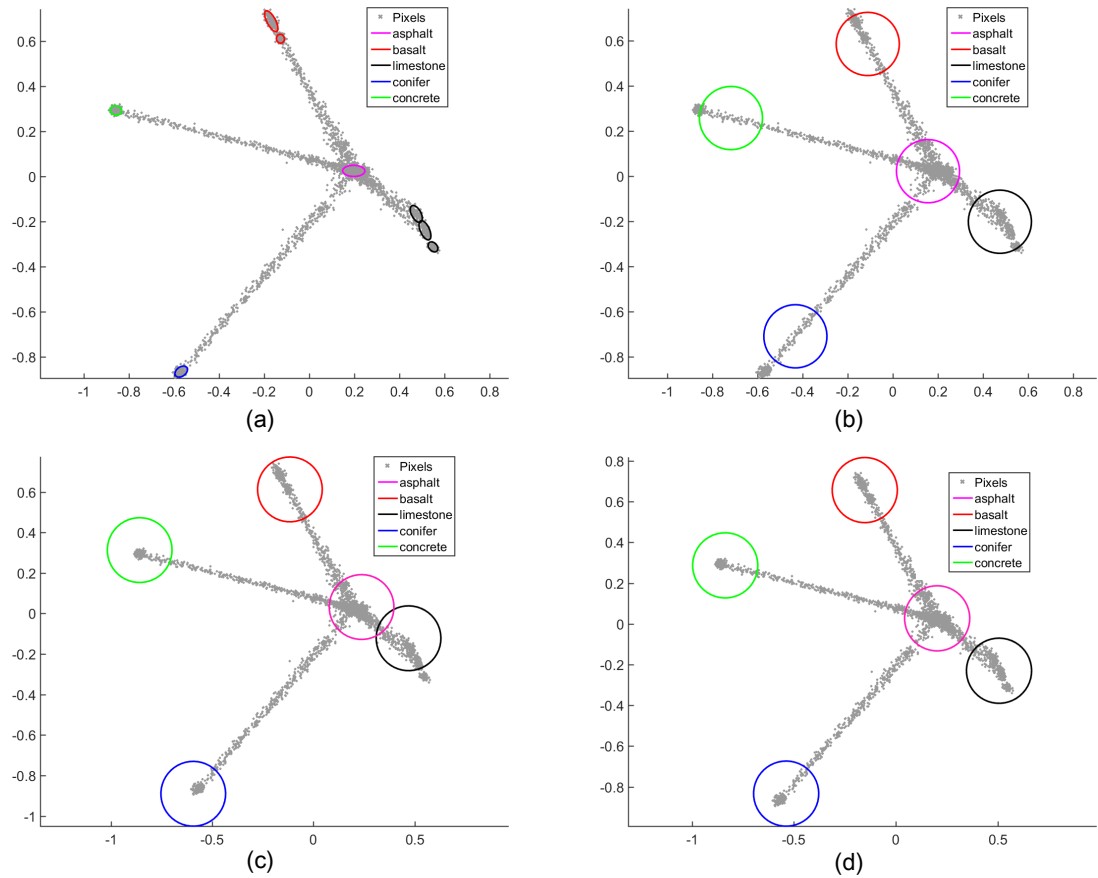

**Figure 14.** The scatter plot of the synthetic dataset under different initial conditions. The gray dots are the pixels when projected by PCA. (**a**) The original endmember scatter plot of the synthetic dataset with estimated GMM. (**b**) The endmember scatter plot by K-means initialization. (**c**) The endmember scatter plot by VCA initialization. (**d**) The endmember scatter plot by region-based VCA initialization.

**Table 7.** Abundance errors for synthetic dataset under different initialization conditions.

| $\times 10^{-4}$ | K-Means | | | VCA | | | Region-Based VCA | | | NCM | BCM |
|---|---|---|---|---|---|---|---|---|---|---|---|
| | **SGSGMM** | **GMM** | **SGSNMF** | **SGSGMM** | **GMM** | **SGSNMF** | **SGSGMM** | **GMM** | **SGSNMF** | | |
| Limestone | **402** | 615 | 818 | **278** | 524 | 668 | **208** | 459 | 672 | 566 | 743 |
| Basalt | **97** | 194 | 330 | **85** | 145 | 262 | **80** | 197 | 261 | 278 | 311 |
| Concrete | **301** | 515 | 533 | **158** | 425 | 448 | **118** | 340 | 463 | 460 | 586 |
| Conifer | 159 | **147** | 254 | **72** | 147 | 173 | **51** | 129 | 175 | 248 | 273 |
| Asphalt | **84** | 190 | 319 | **108** | 192 | 236 | **81** | 161 | 236 | 262 | 277 |
| Whole map | **209** | 332 | 451 | **140** | 287 | 357 | **107** | 257 | 359 | 363 | 438 |

(2) Sensitivity analysis to different size superpixels: Figure 15 shows the performance of SGSGMM with different group sizes. The initialization is the default region-based VCA. From Figure 15e, we can see that when the size of superpixels is set to relatively larger, the precision of its abundance map is declining; further, comparing Table 7, we find that the precision value is gradually approaching to the GMM with the same initialization conditions. This also demonstrates that when the group size is set so large that there only exits one superpixel, the pixels within one spatial group will not be homogeneous and will no longer expect to share the same sparse property. The group sparsity priors will have similar sparsity constraints to the GMM model. When the size of superpixels is set to $w = 7, w = 8$, the performance of the unmixing becomes the best. However, when the group sizes is setting smaller, the accuracy of the unmixing will decrease. This is because when the superpixel is too small, there is not enough data for training, and those superpixels will have no statistical significance.

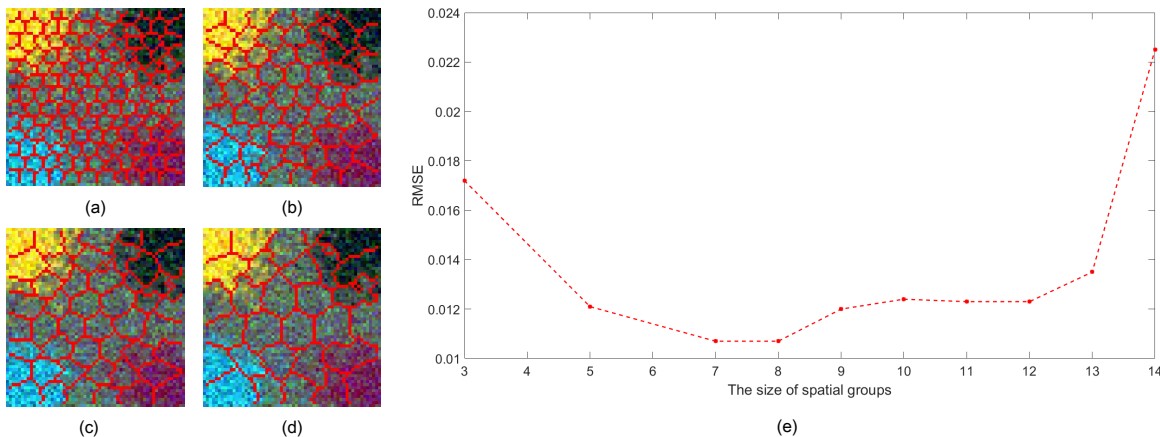

**Figure 15.** Performance analysis of SGSNMF with respect to different size superpixels. (**a**) $w = 5$. (**b**) $w = 7$. (**c**) $w = 9$. (**d**) $w = 11$. (**e**) RMSE of abundance values with respect to $w$.

(3) Efficiency analysis with real hyperspectral data: The efficiency comparison with synthetic and real HSI is provided in Table 8. In this experiment, the original data is reduced to 10 dimensions by implementing PCA. The maximum number of iterations for all the algorithms is set to 100, and the convergence threshold is set is 0.2%. The experiments are performed on the laptop with 2.6-GHz Intel Core CPU, 16GB memory. From Table 8, we can see that the time required for the SGSNMF algorithm is significantly less than other algorithms. As other algorithms are distribution-based methods, more running time is required to search for the endmember combinations. From the detailed procedures in Algorithms 1 and 2, we can find that for SGSGMM, each iteration in the estimation of abundances has spatial complexity $O(|\mathcal{K}|NB^2)$ and time complexity $O(|\mathcal{K}|NB^3)$. NCM has the same complexity but with $|\mathcal{K}| = 1$. Hence, compared with the Gaussian mixture models, less runtime is required. Compared with the GMM and SGSGMM algorithms, we can see that our algorithm requires fewer time resources than GMM. The complexity of these algorithms is the same, and, due to the SLIC segmentation preprocessing, for SGSGMM, the overall running time should be larger. However, the pixels in the same superpixels incline to have common features and a high spatial correlation, which will speed up the convergence of the algorithm in this homogeneous regions. Therefore, the overall convergence time of SGSGMM is relatively small.

**Table 8.** Efficiency comparison for the real hyperspectral data.

| *Times* (*s*) | SGSGMM | GMM | SGSNMF | NCM | BCM |
|---|---|---|---|---|---|
| Gulfport | 1611 | 2293 | **38** | 268 | 1008 |
| Salinas-A | 803 | 980 | **15** | 528 | 2017 |

Limitation: As seen from the above complexity analysis, for the GMM and SGSGMM algorithms, the computational resources needed are relatively large. The main factors affecting the efficiency of GMMs are $|\mathcal{K}|$ and $B$. When the number of components, $K_j$, increases, the complexity would grow exponentially. In the actual processing, we can introduce the thresholds to reduce the number of components or reducing the number of pure pixels to a fixed number by random sampling. Another limitation is that our method is only tested in the scene with a large number of pure pixels region. As our method is a statistical learning method, in the highly mixed scenes, such as spaceborne data, the presence of regions of pure pixels may not hold, and the pure pixel samples are not sufficient for training to model the continuous endmember distribution. Note that this limitation exists more or less in the method based on the statistical analysis. On the basis of ensuring the feasibility, and to verify the performance of the proposed method when considering the mixed pixel problem, the synthetic experiment is designed with adequate mixed pixels in the dataset together with enough pure pixel samples.

## 6. Conclusions

In this paper, a novel unmixing algorithm based on Gaussian mixture model (GMM) and spatial group sparsity constraint is proposed. In an attempt to solve the problems caused by endmember variability and fully exploit the possible spatial correlation, we adopt SLIC segmentation to generate the spatial groups and cut the HSI into different nonoverlapping regions. In these regions, pixels are highly spatial correlated. The mixing pixel and its associated abundance within a local spatial group should share the same prior property constraints. Thus, under the Bayesian framework, we put the spatial prior and the sparsity of the abundance as a modified mixed-norm regularization into the objective function as prior knowledge. Experiments on both simulated and real hyperspectral data demonstrate that the proposed algorithm can achieve higher precision unmixing results compared with other state-of-art methods.

**Author Contributions:** Conceptualization, Q.J. and X.M.; Funding acquisition, Y.M.; Methodology, Q.J.; Resources, Q.J.; Software, Q.J.; Supervision, Y.M., E.P., C.S., F.F., and J.H. and H.L.; Writing—original draft, Q.J.; Writing—review editing, Q.J. and X.M.

**Funding:** This work was supported by the National Natural Science Foundation of China under Grant nos. 61805181, 61773295, 61601397, and 61903279.

**Conflicts of Interest:** The authors declare no conflicts of interest.

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
