# Peer review of "Hyperspectral Unmixing with Gaussian Mixture Model and Spatial Group Sparsity"

_remotesensing, doi:10.3390/rs11202434_

Round 1

Reviewer 1 Report

This paper presents a method called spatial group sparsity constraint based on Gaussian mixture model (SGSGMM) to perform hyperspectral unmixing. It utilizes a modified simple linear iterative clustering (SLIC) algorithm to first perform superpixel segmentation on a PCA-dimension-reduced version of the data. This version is used as initialization for the main algorithm.

The authors then introduce a density function that rewards spatial smoothness in the pixel abundance assignments as well as sparsity. Individual pixels are modeled as a linear combination of endmembers, which follows a mixture of Gaussians (allowing for variability). Lastly, the EM algorithm is utilized to optimize the density function for specific data. 

The approach and presentation are logical and intuitive. I have some comments and questions as to some of the specific results, that I hope to have answered before I can recommend publishing. These comments follow.

----------------------------------------------------------------------

General comments:

1. Line numbers are inconsistently available throughout the document. It would be helpful to make sure that they are displayed on each line in the next draft.

2. There are some grammar and spelling mistakes that make readability challenging at some points.

3. I believe the paper could benefit from a slight expansion in the discussion section. Some particular questions I have are:

a) How do the spatial and spectral resolution of the hyperspectral imagery would affect the algorithm? I would expect that at least the superpixel size parameter w should depend a great deal on the spatial resolution.

b) How sensitive is SGSGMM to the superpixel segmentation initialization?

c) A short comparison of the different types of scenes would be helpful. Does the algorithm behave differently, for example, in a scene with more vegetation vs. more minerals?

d) Would a similar approach work for scenes with a lot of water, which is more nonlinear in its mixing?

More specific comments:

4. Is the sum-to-one constraint necessary? This condition seems to assume that the illumination of all the pixels (and the endmembers) is

5. In line 99 - the first word is "property." Should this be "probability"?

6. After line 166, the beginning of the next paragraph states that the entire HSI cube cannot be used for superpixel segmentation. It might be better to state that this applies to the method that is used for this paper, rather than making it a general statement about the

7. In line 171, what is x_ik? Is this the reflectance value of band k at pixel x_i?

8. In line 266, the text states "discrete and sparse" points are not well estimated. By sparse, does this mean "spatially isolated"? If so, it might be helpful to use change to this term, since "sparse" appears to refer to a different property in most of the text. If "sparse" was actually intended, then I am confused by the text description that follows.

9. Figure 3 - the endmember signatures and the background don't show up very well on a printed copy. This is especially the case when there are multiple components. I suggest that the endmember signature curve be thickened and the gray background be made lighter.

10. In Figure 3, it also appears that different components have a nearly identical signature; does the GMM provide a significant benefit over a single Gaussian then?

11. In Figure 5 (and Figures 9, 13), the x-axis has significant bins that are negative. If I'm understanding these histograms correctly, they are the probability/frequency that the algorithm assigns the given abundance value to the pixel. So for example "basalt" has a significant number of pixels that are assigned to be -0.1 basalt. Is this interpretation correct? If so, what happened to the nonnegativity constraint? If my interpretation of these histograms is incorrect, please clarify the captions to make the plots more understandable.

12. In Figures 7, 11, I suggest removing all the single colorbars and replacing them with just one colorbar on the side. The individual colorbars end up taking up a lot of space and don't really add to the presentation.

13. Figure 14 - I'm not sure if I'm understanding this figure correctly. Is this figure plotting the 2nd principal component vs. the 1st principal component for all the pixels in the image? How are the different types of material (e.g. asphalt) found? Are they manually circled, or was an algorithm used to determine each region? Why aren't all pixels assigned to a given (dominant) material?

Reviewer 2 Report

The authors build up on the paper by Zhou et al. from Paul Gader's group, modelling uncertainties due to endmember variabilities with Gaussian Mixture Models (available at https://arxiv.org/pdf/1710.00075.pdf) . They do so by introducing a stronger spatial prior based on sparsity and a previous segmentation step, carried out on the first PC of the image, and yielding in output super pixels.

The idea is interesting, but there are several points to be fixed before the advantages of the proposed method can be assessed in a fair and convincing way. I mention the issues I came across in no specific order.

Additional efforts required to use the algorithm in terms of parameters setting and running time. Some sensitivity analysis is carried out in Section 5, but: What about the computational resources needed by the algorithm? Running times should be reported for the proposed algorithm and the competitors. And how would these scale for larger images? What about the initialization of smooth and sparsity constraints \beta_1 and \beta2_? What about the parameter w for scenes of different scales or spatial resolution? The experiments on the synthetic dataset are not realistic, as they use spectra in the range 0.4 - 14 micrometers. Usually HS sensors are in the range [0.4-1], [0.4-2.5] or [8-13] micrometers. Also [Zhou et al.] use a similar dataset, but in a relevant spectral range. This should be adjusted. The English has several typoes and several passages were hard to understand for me. For example lines 157-160, describing the limitation of the GMM-based algorithm, did not result clear to me, and I missed the message that the authors want to get through. In figure 6 the labels grass and roof are swapped. The experiment on the Mississippi Gulf dataset appears to be conducted in an unclear way, and it is for me the main issue with this paper. The authors use the same dataset as in [Zhou et al.], and claim that they are using the same ROI, too. But the ROIs in the 2 papers are different, see for example the extent of the roof class (incorrectly labeled as grass in Fig. 6), which includes obvious pixels in the SW corner which do not belong to the class roof. The feeling is that such pixels are added to the ground truth to better match the abundance maps in Fig. 7. The authors (Table 3) report an average abundance error for the previous GMM model of 0.14, outperformed by the proposed method (0.082). But in [Zhou et al.] the error is 0.056. This shows that: The claim that the proposed method outperforms [Zhou et al.] should be reviewed Probably GMM was applied with suboptimal parameters, or due to the weaker spatial prior was more affected by the the different lableing of the ROI. This automatically makes the Salinas A experiment dubious (also because that dataset is usually quite easy to process, exception made for the bimodal class "corn / senesced weeds" for which SGSNMF, NCM and BCM fail) Right now hyperspectral sensors are being launched in space (DESIS, HISUI, PRISMA...). How would this method work if applied to spaceborne images at 30 m resolution, where a segmentation step is often not meaningful or possible, given the predomincance of mixed pixels? Is this method only recommended for high resolution airborne HS images?

To wrap it up, the idea is interesting but the experimental section is not convincing for several reasons, and it must be in order to justify the additional efforts (parameters, processing steps) required from the proposed algorithm. Furthermore, applicability to present and future spaceborne data (which will be the main source of HS data for the scientific community) should be at least discussed. The English should be revised in order to assess the mentioned points in a convincing way.

Round 2

Reviewer 2 Report

Thanks for fixing some points and convincingly clarifying the others issues I raised. Now the paper really shows the advantages of the proposed approach. My bad for raising a non-existing problem such as the increased spectral range of the synthetic dataset. The only problem I still see is the English editing for several sentences (e.g. section Experiment Result --> Experimental Results).

A last suggestion is to specify explicitly that results differ from Zhuo, Gader, .. paper as a different ROI and evaluation metric have been used: readers could be concerned for this as I was, and would not have the response to this reviewer available. I do not need to see a revised version of the manuscript.
